# Conceptor-Aided Debiasing of Large Language Models

Li S. Yifei[1], Lyle Ungar[1], João Sedoc[2],

[1]University of Pennsylvania, [2]New York University
{liyifei, ungar}@upenn.edu, jsedoc@stern.nyu.edu

## Abstract

Pre-trained large language models (LLMs) reflect the inherent social biases of their training corpus. Many methods have been proposed to mitigate this issue, but they often fail to debias or they sacrifice model accuracy. We use *conceptors*–a soft projection method–to identify and remove the bias subspace in LLMs such as BERT and GPT. We propose two methods of applying conceptors (1) bias subspace projection by post-processing by the conceptor NOT operation; and (2) a new architecture, conceptor-intervened BERT (CI-BERT), which explicitly incorporates the conceptor projection into all layers during training. We find that conceptor post-processing achieves state-of-the-art (SoTA) debiasing results while maintaining LLMs' performance on the GLUE benchmark. Further, it is robust in various scenarios and can mitigate intersectional bias efficiently by its AND operation on the existing bias subspaces. Although CI-BERT's training takes all layers' bias into account and can beat its post-processing counterpart in bias mitigation, CI-BERT reduces the language model accuracy. We also show the importance of carefully constructing the bias subspace. The best results are obtained by removing outliers from the list of biased words, combining them (via the OR operation), and computing their embeddings using the sentences from a cleaner corpus.[1]

## 1 Introduction

LLMs such as BERT (Devlin et al., 2019) and GPT (Radford et al., 2019; Brown et al., 2020) are extremely successful in most natural language processing (NLP) tasks. However, since they are trained on texts written by humans, the social bias is inherited and represented in the parameters of LLMs (Bolukbasi et al., 2016; Caliskan et al., 2022). For example, gender bias has been found in contextualized embeddings (May et al., 2019; Zhao

---

[1]Code link: https://github.com/realliyifei/conceptor-debias-llm.

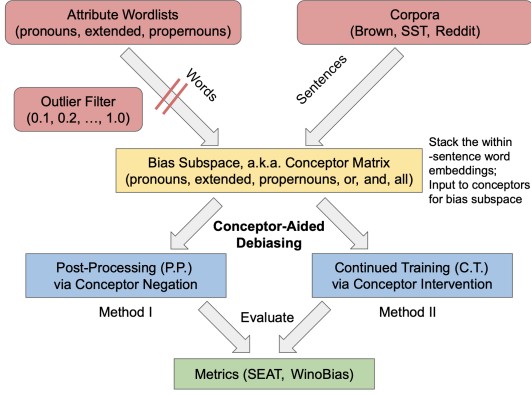

Figure 1: The pipeline of the conceptor-aided debiasing paradigm. We first use different settings (wordlists with outlier filter and corpora) to generate the best bias subspace (conceptor matrix), then apply them to two conceptor-aided debiasing methods and measure the debiasing performance by two evaluation metrics. The experiment is conducted on two LLMs: BERT and GPT.

et al., 2019). Therefore, many researchers have developed debiasing techniques to improve the social fairness of NLP. However, such debiasing often fails to debias effectively and reduces language model performance in downstream tasks (Meade et al., 2022). Furthermore, most debiasing methods neither follow Bommasani et al. (2020)'s suggestion to reduce bias in all layers nor tackle intersectional bias in an efficient way (Lalor et al., 2022).

In this paper, we challenge Karve et al. (2019)'s conclusion that *conceptor negation* fails to debias BERT stably. Instead, we are the first ones to empirically find that as a soft shrinkage of the principal components of the subspace defined by the list of biased words (Liu et al., 2018), conceptors is a powerful tool to debias LLMs such as BERT and GPT using either post-processing or continued-training. In this process, we demonstrate the effect on debiasing performance of choosing different corpora, subspace removal methods, and criteria for selecting the list of bias attribute words that are used to

construct the bias subspace. Further, we unprecedentedly show that the conceptor can tackle varied types of biases (e.g. gender, race) intersectionally and efficiently by its unique logical operation.

Specifically, the *attribute wordlists* at the core of our method, and the methods we build on, are sets of attribute words related to bias. These typically come in opposing pairs (e.g. 'man'/'woman', 'prince'/'princess'). Bolukbasi et al. (2016), Liang et al. (2020) and others use the first principal component (PC) to define the *bias subspace*–which can be later subtracted entirely to debias. We similarly construct such subspaces, but use conceptors as a 'soft' way to remove them–downscale the PC adjusted by a regularized identity map. When generating such wordlists, it may be more representative of bias by removing outliers in the embedding space. Considering the embeddings are contextualized, we select the contextualized token-level word embeddings using sentences from a specific *corpus*. Then we stack them to generate a bias subspace in a form of a conceptor matrix for the debiasing in the next step. The pipeline is shown in Figure 1.

This work contributes the following:

- Employs *conceptor negation post-processing* to debias LLMs such as BERT and GPT, beating most SoTA while retaining useful semantics and robustness in multiple scenarios
- Explores *conceptor-intervened BERT (CI-BERT)*–a novel model architecture that continues training BERT after incorporating conceptors within all of BERT's layers
- Illustrates how different corpora, bias attribute wordlists, and outlier removal criteria impact debiasing performance
- Demonstrates conceptor-aided methods can be generalized to different layers of LLMs and various types of biases and can mitigate them intersectionally by its unique logical operation

## 2 Related Work

### 2.1 Bias Manifestation

Multiple demographic biases are common in society. Among them, gender bias is the most well-studied in academia, given its omnipresence and bi-polarity (Bolukbasi et al., 2016; May et al., 2019; Kurita et al., 2019). Other social biases (e.g. racial, religious) are also widespread in LLMs and attracting increasing attention (Nangia et al., 2020; Nadeem et al., 2021; Meade et al., 2022).

Such social bias manifests itself in all layers of the contextualized representations of LLMs like BERT and GPT (Bommasani et al., 2020); and Kaneko and Bollegala (2021) show that debiasing all layers is more effective. Moreover, Lalor et al. (2022) indicates the importance of addressing varied biases in different dimensions. Thus, a new challenge is raised on how to adapt current methods or develop novel paradigms to mitigate the bias in each layer and across multiple social dimensions.

### 2.2 Debiasing Techniques and Challenges

We collect the mainstream SoTA debiasing methods (Overview: Meade et al. (2022); Xie and Lukasiewicz (2023)), each with typical examples:

(1) Bias Subspace Projection (BSP): the classic method of bias subspace subtraction is to first capture the bias subspace determined by attribute words in the corpora and then project the bias direction out from the language embeddings. This can be done by post-processing as either hard projection (Bolukbasi et al., 2016; SENTENCEDEBIAS, Liang et al., 2020) or soft projection (Karve et al., 2019). Some variants attain a similar goal by training a linear classifier (INLP, Ravfogel et al., 2020) or fine-tuning LLMs (Kaneko and Bollegala, 2021).

(2) Counterfactual Data Augmentation (CDA): swapping the bi-polar bias attribute words (e.g. her/him) to rebalance the training dataset and therefore decrease the gender bias (Webster et al., 2020; Barikeri et al., 2021).

(3) Dropout Regularization (DROPOUT): in combination with an additional pre-training, increasing the dropout components inside the transformer-based language models can lead to lower bias (Webster et al., 2020).

(4) SELF-DEBIAS: by using specific templates to encourage LLMs to generate toxic output and then modifying the original output distribution of the model by a decoding algorithm, Schick et al. (2021) makes use of the internal knowledge of language model to debias in a post-hoc manner.

Further, it is common to combine multiple such methods. For instance, Zhao et al. (2019) and Liang et al. (2020) combine the techniques of data augmentation and hard debiasing. However, per the discussion in Meade et al. (2022), the methods often neither debias as well as they claim (e.g. CDA, DROPOUT, SENTENCEDEBIAS), nor do they maintain the model's capability for downstream tasks (e.g. CDA, DROPOUT, INLP). Worse, some techniques like CDA and DROPOUT increase the bias

measured on SEAT–a test of language bias which we will describe in Section 5. This dilemma challenges us to develop new methods to further reduce bias while retaining meaningful semantics. Last, the majority of debiasing methods ground the bias by word list; different lists can lead to different debias performance (Antoniak and Mimno, 2021).

## 2.3 Conceptors in NLP

Conceptors–a soft projection method supporting conceptual abstraction and logical operations (Jaeger, 2014)–has been adapted to NLP domains such as debiasing (Liu et al., 2018; Sedoc and Ungar, 2019; Karve et al., 2019), continual learning (Liu et al., 2019a), and semantic information enrichment (Liu et al., 2019b). *Conceptor negation* is a soft shrinkage of the PCs of a subspace such as stop words or, in our case, of the target words defining the bias directions (Liu et al., 2018). Therefore it has the potential to debias better than hard projection (e.g., Bolukbasi et al., 2016) while retaining enough semantics. Mathematically, it can capture, conjoin, and negate the bias concepts by logical operation, and thus can **deal with intersectional bias efficiently**.

Although Karve et al. (2019) showed that debiasing conceptors can successfully debias both static embeddings such as Glove, Word2vec, and Fasttext, and contextual embeddings such as ELMo (Peters et al., 2018), they state that the performance in BERT is far less consistent and effective than other word representations. We discover that this is the result of their having selected the wrong set of attribute words, which leads to a poor bias subspace.[2] Another difference is that the BERT tokens of attribute words should be averaged if they contain multiple subwords after tokenization (Liang et al., 2020; Kaneko and Bollegala, 2021).

## 3 The Mechanism of Conceptors

Let us take a closer look at the mathematics of conceptors: considering a set of vectors $\{x_1, \cdots, x_n\}$, $x_i \in \mathbb{R}^N$ for all $i \in \{1, \cdots, n\}$, a conceptor matrix $C$ is a regularized identity map that minimizes

$$\frac{1}{n}\sum_{i=1}^{n}\|x_i - Cx_i\|_2^2 + \alpha^{-2}\|C\|_F^2 \qquad (1)$$

---

[2]We fixed the coding issues.

where $\|\cdot\|_F$ is the Frobenius norm and $\alpha^{-2}$ is a scalar hyper-parameter called *aperture* [3]. It can be shown that $C$ has a closed-form solution:

$$C = \frac{1}{n}XX^\top\left(\frac{1}{n}XX^\top + \alpha^{-2}I\right)^{-1} \qquad (2)$$

where $X = [x_i]_{i\in\{1,\cdots,n\}}$ is a data collection matrix whose $i$-th column is $x_i$. Intuitively, $C$ is a soft projection matrix on the linear subspace where the typical components of $x_i$ samples lie so that it can capture the components that all representations roughly share. Therefore, different from PCA projection which removes the first several principal components (PCs) completely, the conceptors method softly downscales the PCs adjusted by a regularized identity map (Figure 2).

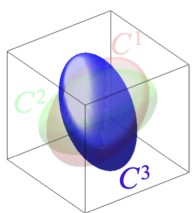

Figure 2: Geometry of three conceptors in the shape of ellipsoids (Jaeger, 2014).

Conceptors support Boolean operations such as **NOT** ($\neg$), **AND** ($\wedge$) and **OR** ($\vee$). For two arbitrary conceptors $C_1$ and $C_2$, we have

$$\neg C_1 = I - C_1 \qquad (3)$$
$$C_1 \wedge C_2 = (C_1^{-1} + C_2^{-1} - I)^{-1} \qquad (4)$$
$$C_1 \vee C_2 = \neg(\neg C_1 \wedge \neg C_2) \qquad (5)$$
$$= I - ((I - C_1)^{-1} + (I - C_2)^{-1} - I)^{-1}$$

These logical operations are feasible if $C_1$ and $C_2$ are created by the sets of equal sizes (Jaeger, 2014), as shown in Figure 3. This reveals the potential for debiasing by combining different conceptors learned from different bias subspaces. This is helpful both in combining different wordlists for the same bias (e.g. gender) or different wordlists for different protected classes (e.g. gender and race).

## 4 Debiasing Sentence Representations

### 4.1 Bias Subspace Setting

We explore the impact of different choices of attribute wordlists, the corpora used to find their embeddings, and how the wordlists are combined and filtered to remove outliers, on the quality of bias

---

[3]The default value of $\alpha$ is 1; we empirically find that gridsearching is not helpful for debiasing so keep it as default

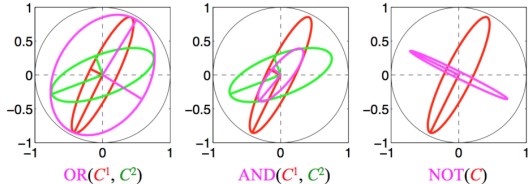

Figure 3: Visualizing the boolean operations on two conceptor matrices. The OR (AND) operator leads to a conceptor matrix (in pink color) with the smallest (largest) ellipsoid (He and Jaeger, 2018). In our case, it is then negated by the NOT operator to debias.

subspace, and hence the debiasing (Fig 1). Different procedures of bias subspace construction yield significantly different debiasing performances.

**Corpora**  We compare three corpora: (1) the **Brown** Corpus (Francis and Kucera, 1979), a collection of text samples of mixed genres; (2) the Stanford Sentiment Treebank (**SST**; Socher et al., 2013), a polarized dataset of 10,662 movie reviews; and (3) a **Reddit** Corpus (Liang et al., 2020), a dataset collected from discussion forums about relationships, electronics, and politics. The reason is to see how the language formality and topic breadth of texts impact the debiasing, the Brown corpus is formal and contains 15 genres, the Reddit corpus is informal with 3 domains and the SST corpus is informal, has only one domain. They are used to provide embeddings for the attribute words.

**Combining and Filtering Wordlists**  We compare five ways of using three different wordlists to create conceptor bias subspaces.

The three wordlists are gender words originating from different sources: the *pronouns wordlist* is a set of common terms that are specific to particular genders, such as 'daughter' or 'son'; the *extended wordlist*, an extension of the former, contains less frequent words such as 'cowgirls' or 'fiancees'; and *propernouns wordlist* is comprised of proper nouns like 'Tawsha', 'Emylee', and so on.

There are five methods of using these three wordlists to generate a bias subspace. We can use each of them individually (their subspaces are named the same as themselves: **pronouns**, **extended**, and **propernouns**, respectively). We can also combine them in two ways: either by concatenating them as a single list generating a corresponding subspace (named **all**); or by running the conceptor OR operation–a Boolean operation of conceptors described in  subsection 2.3–on the

three corresponding conceptor matrices to generate what can be viewed as a union of the three bias subspaces (named **or**).

Unlike Karve et al. (2019), to study the effects of removing outliers from the wordlists, we first project the LLM's embeddings of the words in the wordlist to a 2-dimensional UMAP clustering (McInnes et al., 2018) space, shown in Figure 4, and then filter the outliers by percentile on their $(x, y)$-coordinate. The outliers are defined as the points that fall outside of 1.5 times the inter-range (IR), which is the difference between $p$-th and $(1-p)$-th percentile. We iterate $p$ from 0.1 to 1.0 with step size 0.1 to generate different wordlists and then test how well each debiases. Our goals are to detect the negative effect of outliers on debiasing performance and to explore which percentile here is optimal for debiasing.

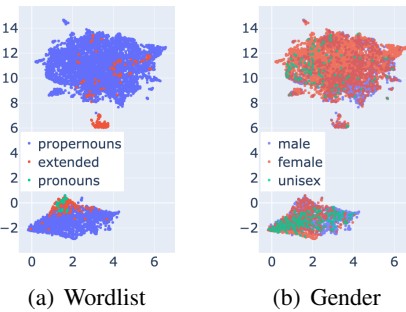

(a) Wordlist  (b) Gender

Figure 4: 2D UMAP BERT Embeddings of Words.

## 4.2  Debiasing Methods

We propose and explore two kinds of conceptor-aided debiasing: *conceptor post-processing*, and *conceptor-intervened continued training*. They are abbreviated as P.P. and C.T. respectively in tables.

**Conceptor Bias Subspace Construction**  We construct the conceptor negation matrix $\neg C$ as demonstrated in Algorithm 1, where matrix $X$ is a stack of the within-sentence contextualized embeddings of the words. The words are determined by attribute wordlists and the sentences are from the specified corpus as mentioned in Section 4.1. Note that we do not need the "difference space" of bipolar bias as the conceptor projection matrix is applied to the original space–in this way the conceptor method is different from the so-called hard-debiasing (Bolukbasi et al., 2016). To ensure contextualization we remove the less-than-four-word sentences. Also, following Kaneko and Bollegala (2021)'s idea, if a word crosses multiple sub-tokens,

then its contextualized embedding is computed by averaging the contextualized embeddings of its constituent sub-tokens, which is different than the previous conceptor works.

**Conceptor Negation and Post-Processing** Next, we post-process the sentence embeddings $t$ which contain attribute words and target words, by taking the matrix product of $\neg C$ to subtract the bias subspace, rendering debiased embeddings $t^*$, as demonstrated in the last part of Algorithm 1. Each BERT layer manifests different levels of bias (Bommasani et al., 2020). To maximize the effectiveness of $\neg C$, we want $\neg C$ to be generated from the corresponding layer. Therefore, we are the first ones to test the debiasing performance by using different conceptor matrices generated by different layers of the language model and to explore whether conceptor post-processing generalizes well on each layer of LLMs and on different LLMs (BERT and GPT).

**Intersectioanl Debiasing** Importantly, not only can conceptors mitigate different types of biases such as gender and race respectively, but it can also conjoin and negate these biased concepts together due to its magical logical operations. It is natural that societal biases co-exist in multi-dimensions: such as "African Male" rather than "African" and "Male". Therefore, it is efficient that conceptors can tackle them intersectionally by utilizing the previously constructed bias subspaces via its OR operation to construct the new mixed conceptors.

**Conceptor Intervention and Continued Training** The varying levels of bias across BERT layers suggest the possible utility of an alternate approach to mitigate the bias. Accordingly, we construct a new architecture, *Conceptor-Intervened BERT* (CI-BERT), by placing the corresponding conceptor matrix after each layer of BERT (Figure 5). We then continue training the entire model to incorporate the model weights with the bias negation captured by the conceptors in each layer. Thus we can take the biases in all layers into account so that we can mitigate the layerwise bias simultaneously.

CI-BERT architecture can be used in three ways. We can load the original pre-trained weights to CI-BERT and directly render the language embeddings (**Type I**; CI-BERT $\times$ original weights). Alternatively, we can continue training the model using CI-BERT to get newly trained weights; then we can load these weights back to either the original off-the-shelf BERT architecture (**Type II**; BERT $\times$ trained weights) or to the new architecture CI-BERT (**Type III**; CI-BERT $\times$ trained weights).

# 5 Quantifying Bias

## 5.1 Sentence Encoder Association Test

The Sentence Encoder Association Test (SEAT) (May et al., 2019) is an extension of the Word Encoder Association Test (WEAT) (Caliskan et al., 2017). It can measure the bias at the sentence level in different kinds of bias (Meade et al., 2022).

SEAT uses two types of words: *attribute* words $\mathcal{W}_a$ (e.g. he/she) and *target* words $\mathcal{W}_t$ (e.g. occupations), which we expect to be gender-neutral. That is, the associations between $w_a/w_a' \in \mathcal{W}_a$ and $w_t \in \mathcal{W}_t$ should be no difference in the sentence-template representations of LLMs.

Denote the sentence sets of attribute words as $A$ and $A'$, and of target words as $T$ and $T'$, we have:

$$c(A, A', T, T') = \sum_{t \in T} c(t, A, A') - \sum_{t' \in T'} c(t', A, A')$$

where for each sentence $s$, we have $c(s, A, A')$, the difference of the mean cosine similarity of $s$ concerning sentences from between $A$ and $A'$; as

$$c(s, A, A') = \frac{1}{|A|} \sum_{a \in A} \cos(s, a) - \frac{1}{|A'|} \sum_{a' \in A'} \cos(s, a')$$

The amount of bias is given by the effect size

$$d = \frac{\mu\left(\{c(t, A, A')\}_{t \in T}\right) - \mu\left(\{c(t', A, A')\}_{t' \in T'}\right)}{\sigma\left(\{c(a, T, T')\}_{a \in A \cup A'}\right)}$$

where $\mu$ and $\sigma$ denote the mean and standard deviation, respectively. The smaller the absolute value of $d$ is, the less bias has been detected. The one-sided p-value measures the likelihood that a random resampling of the sentence set that contains attribute words would generate the observed test statistic.

## 5.2 Gender Co-Reference Resolution

As described by Gonen and Goldberg (2019), SEAT can detect only the presence but not the absence of bias. To further understand how the conceptor-aided methods work on debiasing, we adopt an end-task: gender co-reference resolution.

WinoBias (Zhao et al., 2018) provides gender-balanced co-reference tests to evaluate LLMs' neutrality towards pronouns referring to occupations. The tests include pro-stereotypical (PRO) scenarios, where gender pronouns match gender-conforming occupations (e.g., her/nurse), and anti-stereotypical (ANTI) scenarios, where gender pronouns apply to disfavored occupations. The bias is measured by the average and absolute difference in F1 scores between the PRO and ANTI subsets.

**Algorithm 1** CONCEPTOR-DEBIAS: a conceptor-aided post-process algorithm for debiasing LLMs.

---

**Require:** large language model $\mathbb{M}_\theta$ (with parameters $\theta$), bias attribute wordlist $\mathcal{W}$, and corpus $\mathcal{S}$.
1: $X \leftarrow [\ ]$
2: **for** each word $w \in \mathcal{W}$ **do**
3:     **for** each sentence $s \in \mathcal{S}$ **do**
4:         **if** $w$ inside $s$ **then**
5:             $w_c \leftarrow$ the embedding of $w$ inside $\mathbb{M}_\theta(s)$     // get contextualized word embedding
6:             $X \leftarrow X + w_c$     // stack as a matrix
7:         **end if**
8:     **end for**
9: **end for**
10: $C \leftarrow XX^\top(XX^\top + I)^{-1}$     // construct conceptor bias subspace
                                          // note that different $X_i$ yields different $C_i$ for arbitrary $i$
11: $C \leftarrow (C_1^{-1} + C_2^{-1} - I)^{-1}$     // cross bias subspaces by AND operator (if intersectional debias)
12: $C \leftarrow I - ((I - C_1)^{-1} + (I - C_2)^{-1} - I)^{-1}$     // unite bias subspaces by OR operator (for robust debias)
13: $\neg C \leftarrow I - C$     // make negation conceptor matrix by NOT operator
14: **for** each new sentence $t$ **do**
15:     $t^* \leftarrow \neg C \cdot \mathbb{M}_\theta(t)$     // debias sentence by projection
16: **end for**

---

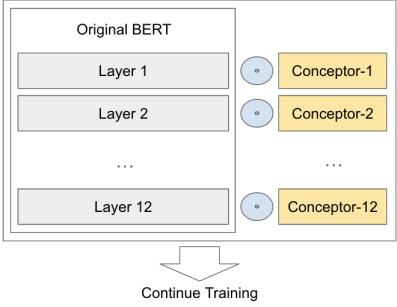

Figure 5: Conceptor-Intervened BERT (CI-BERT). Each model's layer takes the matrix product (blue circle) of the conceptor-X generated from the corresponding layer X. It can be used directly or continually trained.

Based on this, de Vassimon Manela et al. (2021) develop two intuitive metrics, *skew* and *stereotype*, to better probe model fairness. In formula,

$$\mu_{\text{Skew}} \triangleq \frac{\left| \text{F1}_{\text{pro}}^M - \text{F1}_{\text{pro}}^F \right| + \left| \text{F1}_{\text{anti}}^M - \text{F1}_{\text{anti}}^F \right|}{2}$$

$$\mu_{\text{Stereo}} \triangleq \frac{\left| \text{F1}_{\text{pro}}^M - \text{F1}_{\text{anti}}^M \right| + \left| \text{F1}_{\text{pro}}^F - \text{F1}_{\text{anti}}^F \right|}{2}$$

where superscripts $M$ and $F$ denote male and female respectively and F1 stands for the F1-score. It is shown that there is an approximate trade-off between these two biases. The authors argue that the T2 test set of WinoBias is better than the T1 test set at revealing bias, as the latter is less ambiguous to LLMs. Therefore, we only report T2 here.

## 6 Debiasing Results

This section aims to answer these questions:
- What is the best setting for bias subspace generation within conceptor-aided debiasing?
- Given the best setting, can the conceptor post-processing mitigate bias and beat SoTA?
- Does embedding conceptors into LLMs via continued training beat post-processing?
- What roles can conceptors operators–NOT, OR, AND–play in the debiasing pipeline?

To help comparison, the SoTA debiasing results from Meade et al. (2022) is included in the tables.

### 6.1 Models

To investigate the generalization of conceptor debiasing, we explored different scales of typical LLM families, which are: BERT-T (*bert-tiny*), BERT (*bert-base-uncased*), BERT-L (*bert-large-uncased*), GPT2 (*gpt2*), GPT-L (*gpt2-large*), and GPT-J (*gpt-j*). We did not test on GPT3 and Chat-GPT since their embedding models (e.g. *text-embedding-ada-002*) do not support the contextualized embedding on token level. However, due to the similar modeling, once we have such embedding, conceptor techniques can be transferred.

### 6.2 Bias Subspace Construction with Robustness Boosted via OR Operator

We construct the conceptor bias subspaces upon the different combinations of corpora, wordlist selections, and outlier removal.

To evaluate corpora, by testing on the last layer of the BERT, we compare the debiasing result of three different corpora: Brown, SST, and Reddit on SEAT. Table 8 shows that Brown stably provides the best debiasing result even if using different wordlist subspaces. The SST corpus is a close second, while Reddit is by far the worst. The style of the Reddit corpus is most likely least similar to that of the SEAT evaluations.

To evaluate alternate methods of constructing the bias wordlist subspace, we use the five subspaces described in Section 4.1. Among them, the *or* subspace is the most robust; see Table 9, 10 and 11. Combining the *pronouns*, *extended* and *propernouns* subspaces with *or* represents the distinct yet union concepts (and hence subspaces) of each of the wordlists, thus both outperforming individual wordlists and outperforming the *all* subspace, which simply concatenates all the wordlists, giving a less precise subspace definition.

To evaluate wordlist outlier removal, we define the outliers by the UMAP filter as discussed in section 4.1 and generate different percentages of the words that are used to capture bias. For example, the *all* subspace has 2071 words within $0.5 - 1.0$ percentile, 2061 in the 0.4 percentile, 1601 in the 0.3 percentile, 430 in the 0.2 percentile, and 82 in the 0.1 percentile (Table 6). We observe that including fewer words often leads to higher debiasing performance, presumably due to the removal of outliers. However, an extremely small percentile, say 0.1, would harm the effectiveness of debiasing because of the inadequate loss being left (Table 9, 10 and 10). Similar results are obtained if using T-SNE (Van der Maaten and Hinton, 2008).

In conclusion, the optimal setting for BERT-T is "sst-0.5-or" (SST; percentile 0.5; *or* subspace); similarly, for BERT is "brown-0.4-or" (Brown; percentile 0.4; *or* subspace). For other models, if not mentioned, it is *default* as "brown-1.0-or". Henceforth, these settings are held for the conceptor debiasing on the models respectively.

### 6.3 Post-Processing Debias via NOT Operator

For general debiasing via conceptor negation postprocessing, the performance is excellent. The SEAT score of BERT decreases from 0.620 to around $0.350 - 0.400$ in Brown Corpus (Table 9), and can be as low as 0.311 if using the setting "brown-0.4-or", outperforming the debiasing result of CDA, DROPOUT and SENTENCEDEBIASE (Table 1). The success of debiasing is further verified by WinoBias (Table 2), where the skew bias drops from 38.3 to 22.3 without any additional fine-tuning. Although the stereotype bias increases, it is not only expected since these two biases are trade-offs but also acceptable, as they now reach a good balance (de Vassimon Manela et al., 2021).

The debiasing conceptors are robust and generalizable, as shown in Table 1, the debias performance

is consistent in different scales of BERT and GPT models. Note that the settings of BERT-L, GPT2-L and GPT-J are not *tuned* (i.e. *default* setting), which means that they can likely reach much lower SEAT scores. Moreover, conceptors can mitigate the bias in almost all scenarios, no matter using which corpus, bias subspace, or wordlist threshold (Table 9, 11 and 10); no matter which LLMs (Table 1, 15, 17, 16 and ,19) ; no matter in which layer (Table 12, 13 and 18); and no matter which type of biases (Table 3, Table 21 and 22).

| Model | F1 Male | | F1 Female | | Bias | |
|---|---|---|---|---|---|---|
| | Pro | Anti | Pro | Anti | Stereo | Skew |
| BERT | 66.4 | 58.9 | 31.8 | 17.0 | 11.2 | 38.3 |
| + Conceptor P.P. | 69.5 | 48.1 | 52.8 | 20.1 | 27.0 | **22.3** |
| + Conceptor C.T. | 41.0 | 39.3 | 57.6 | 56.6 | **4.1** | 17.0 |

Table 2: F1 of skew and stereotype biases in WinoBias.

### 6.4 Intersectional Debias via AND Operator

Table 3 empirically shows that conceptors not only can mitigate the different type of biases, but also can intersect the existing bias subspaces (e.g. gender and race) to create a mixed conceptor matrix in an efficient way and to debias gender and race respectively. Furthermore, for assessing the intersectional debiasing, we employ the I1-I5 intersectional bias test introduced by Tan and Celis (2019). They adapt the SEAT to examine the privilege associated with the combination of being African/European American and being male or female. The results demonstrate that such intersected conceptor formed via the AND operator can effectively reduce multidimensional bias, lowering the SEAT score from 0.673 to 0.434, while its conceptor counterparts focused solely on single-dimensional bias can only reduce the score to 0.613 and 0.635 respectively.

### 6.5 Conceptor-Intervention Debias

We use CI-BERT architecture to continue to train the models to get the new weights. Then we demonstrate the combinations of architectures and weights as an ablation study (Type I, II, and III). Among them, Type III can outperform conceptor post-processing (Table 1), and Type I and II (Table 4). Compared to the SEAT score after postprocessing, Type I can outperform it at each layer of BERT-T but underperform it at most layers of BERT (Table 13 and 18).

In short, using the CI-BERT with the newly trained weights could receive the lowest bias in the model and is promising to beat post-processing. For example, when using the setting "brown-0.4-

| Model | SEAT-6 | SEAT-6b | SEAT-7 | SEAT-7b | SEAT-8 | SEAT-8b | Gender (AAvg.) | |
|---|---|---|---|---|---|---|---|---|
| BERT | 0.931* | 0.090 | -0.124 | 0.937* | 0.783* | 0.858* | | 0.620 |
| + Conceptor P.P. (tuned) | **0.388** | **-0.078** | -0.292 | **0.179** | **0.594*** | 0.335 | ↓0.309 | **0.311** |
| + Conceptor C.T. | **0.227** | 0.426 | -0.341 | **-0.253** | **-0.344** | -0.088 | ↓0.340 | **0.280** |
| + CDA | **0.846*** | 0.186 | -0.278 | 1.342* | 0.831* | **0.849*** | ↑0.120 | 0.722 |
| + DROPOUT | **1.136*** | 0.317 | 0.138 | 1.179* | 0.879* | 0.939* | ↑0.144 | 0.765 |
| + INLP | 0.317 | -0.354 | -0.258 | **0.105** | 0.187 | -0.004 | ↓0.416 | **0.204** |
| + SENTENCEDEBIAS | 0.350 | -0.298 | -0.626 | **0.458*** | 0.413 | 0.462* | ↓0.186 | 0.434 |
| BERT-L | 0.370 | -0.015 | 0.418* | 0.221 | -0.258 | 0.711* | | 0.332 |
| + Conceptor P.P. (default) | **0.197** | -0.206 | 0.064 | **0.065** | -0.371 | **0.337** | ↓0.125 | **0.207** |
| GPT2 | -0.510 | 0.057 | -0.274 | -0.186 | -0.369 | -0.313 | | 0.285 |
| + Conceptor P.P. (tuned) | **0.092** | 0.316 | **-0.001** | 0.064 | **-0.035** | **-0.062** | ↓0.190 | **0.095** |
| GPT2-L | 1.093* | 0.192 | 0.214 | 1.354* | 0.861* | 1.157* | | 0.812 |
| + Conceptor P.P. (default) | **1.055*** | **0.008** | **-0.089** | 1.406* | **0.282** | **0.992*** | ↓0.173 | **0.639** |
| GPT-J | 1.299* | 0.300 | 0.962* | 1.434* | 0.617* | 1.031* | | 0.940 |
| + Conceptor P.P. (default) | **1.184*** | **0.285** | **0.661*** | **1.284*** | **0.558*** | **1.024*** | ↓0.107 | **0.833** |

Table 1: SEAT effect size of gender debiased BERT and GPT model. Effect sizes closer to 0 indicate less biased sentence representations (**bolded value**). Statistically significant effect sizes at $p < 0.01$ are denoted by *. The final column is the average absolute SEAT score of the first six columns. *Default* means using the default setting: brown corpus, no wordlist filtering, and OR subspace; while *tuned* means using the optimal combination of corpus, wordlist percentile, and conceptor bias subspace. P.P. stands for post-processing, while C.T. stands for continued training. The full version is in Appendixes E and G.

| Model | Gender (AAvg.) | | Race (AAvg.) | | SEAT-I1 | SEAT-I2 | SEAT-I3 | SEAT-I4 | SEAT-I5 | Intersect (AAvg.) | |
|---|---|---|---|---|---|---|---|---|---|---|---|
| BERT | | 0.620 | | 0.620 | 0.389* | -0.424 | 1.195* | 0.525* | 0.834* | | 0.673 |
| + Gender Conceptor | ↓0.309 | **0.311** | | N/A | 0.394* | -0.456 | **1.156*** | **0.413*** | **0.755*** | ↓0.060 | **0.613** |
| + Race Conceptor | | N/A | ↓0.043 | **0.577** | 0.394* | -0.456 | **1.156*** | **0.413*** | **0.755*** | ↓0.062 | **0.635** |
| + Intersect Conceptor[§] | ↓0.029 | **0.591** | ↓0.016 | **0.604** | **0.214** | -0.474 | **0.872*** | **0.207** | **0.403*** | ↓0.239 | 0.434 |

Table 3: SEAT effect size of race, gender, and intersectionally debiased BERT model, where the absolute average SEAT score of gender, race, and intersect are across 6, 7, 5 tests, respectively. The full version is in Appendix I.
[§] It indicates the conceptor matrix generated by its negated AND operation of gender conceptor matrix and race conceptor matrix

| Type | CI-BERT (Arch.) | Trained Weights | SEAT |
|---|---|---|---|
| (Orig.) | | | 0.620 |
| I | ✓ | | 0.336 |
| II | | ✓ | 0.592 |
| III | ✓ | ✓ | 0.280 |

Table 4: The ablation study of architecture and weights of CI-BERT evaluated by SEAT (the same as Table 1).

or", the lowest SEAT score is 0.280, beating the post-processing result of 0.311 and more than half of the SoTA methods. This is verified again by gender co-reference resolution in Table 2–in comparison to its post-processing counterpart, CI-BERT continued training lowers both stereotype bias by 22.9 and skew bias by 5.3 from Test Set 2 of Wino-Bias. This is non-trivial since these two biases are a tradeoff and thus generally hard to decrease simultaneously (de Vassimon Manela et al., 2021).

To further study the feasibility and robustness of CI-BERT continued training concerning the model property. We experiment on both BERT-T and BERT and plot the average SEAT curve along with training steps (Figure 6). Both can beat their post-processing counterparts in some steps during the early training stage, and then the bias fluctuates and increases again, perhaps due to the model relearning the bias during continued training, or oversaturating the conceptor bias projections into its weights.

In comparison, the continual-trained CI-BERT can more stably lower the bias in smaller Bert model. We suspect this is related to the model complexity. The debiasing projection of the last layer's conceptor matrix is upon the last hidden state and thus generated transitively from all the prior layers. Currently, we are embedding all layers' conceptor matrices, which may lead to overlapping and redundant debiasing projection from the prior layers.

### 6.6 Maintaining Meaningful Semantics

To understand how conceptor debiasing impacts the downstream natural language understanding (NLU) tasks, the GLUE benchmark (Wang et al., 2018)–comprised of nine different tasks–is used to evaluate the model after debiasing (Table 5). While there seems to be no consensus about the quanti-

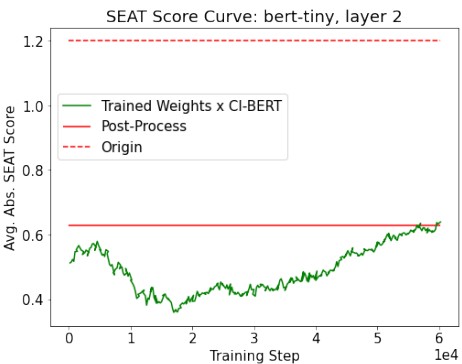

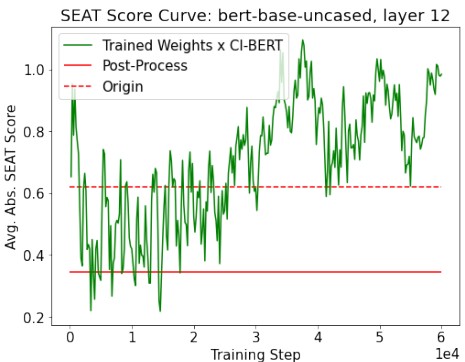

Figure 6: SEAT score curve of CI-BERT continued training. We compare the results with the original embeddings and post-processed embeddings. We test on the last layer of BERT-T (top) and BERT (bottom).

tative threshold of the trade-off between language modeling capability and debiasing performance, a small decrease may be acceptable depending on the downstream tasks. We believe that, in an ideal scenario, the performance on the GLUE benchmark should not significantly decline after debiasing.

The conceptor post-processing of BERT can retain and even improve the useful semantics (increase the average GLUE score by 1.77) for downstream tasks without damage to the model's ability, outperforming any other listed SoTA debiasing methods. Even if scaling to BERT-L, the GLUE is still slightly higher. In comparison, the average GLUE score of conceptor continued-training BERT is relatively low, although it is not the worst among all the methods. This indicates that the continued-training method, while still capable of outperforming its post-processing counterpart under the same setting, may sacrifice NLU abilities.

Since GPT is an autoregressive model, we adopt the SequenceClassification counterpart on the GLUE benchmark, following the method of Meade et al. (2022). The score of GPT2 and GPT-J are decreased slightly by 0.11-0.14, which is an affordable cost, while GPT2-L increases slightly by 0.05.

| Model | | Average |
|---|---|---|
| BERT | | 77.74 |
| + Conceptor P.P. | ↑1.77 | 79.51 |
| + Conceptor C.T. | ↓1.03 | 76.71 |
| + CDA | ↓0.22 | 77.52 |
| + DROPOUT | ↓1.46 | 76.28 |
| + INLP | ↓0.99 | 76.76 |
| + SENTENCEDEBIAS | ↑0.07 | 77.81 |
| BERT-L | | 78.81 |
| + Conceptor P.P. | ↑0.05 | 78.86 |
| GPT2 | | 73.01 |
| + Conceptor P.P. | ↓0.11 | 72.90 |
| GPT2-L | | 75.84 |
| + Conceptor P.P. | ↑0.05 | 75.89 |
| GPT-J | | 78.22 |
| + Conceptor P.P. | ↓0.14 | 78.06 |

Table 5: GLUE validation set results for gender debiased BERT and GPT model. The full version is in Appendixes E and H.

Notice that even when trained on the original BERT architecture, the average GLUE score still drops about 0.3 point. Thus, the lower GLUE score here is not completely caused by CI-BERT, though the actual reason is hard to determine due to training randomness (McCoy et al., 2019).

## 7 Conclusion and Future Work

We have shown that conceptor-aided debiasing can successfully mitigate bias from LLMs (e.g., BERT, GPT) by its NOT operation. Specifically, conceptor post-processing outperforms many state-of-the-art debiasing methods in both debiasing effectiveness and semantic retention. We also tested a new architecture, conceptor-intervened BERT (CI-BERT), which in combination with continued training, takes all layers' biases into account and shows the promise to outperform its post-processing counterpart. However, it might be at the cost of increased instability and worse semantic retention. In all cases, the best conceptor matrices are generally obtained when the bias subspace is constructed using (1) a cleaner corpus, (2) the union of different related wordlists (e.g. pronouns, roles, and names) by the conceptor OR operation, and (3) removal of outliers from the wordlists. We further show that cocneptor-aided debiasing is robust in different LLMs, various layers of models, and varied types of biases. Moreover, conceptors can utilize the current conceptor matrices to construct a new conceptor matrix to mitigate the intersectional bias in an efficient manner by AND operation.

In future research we plan to make CI-BERT and intersectional conceptors more robust and effective.

## Limitations

We list several limitations of our work below.

**1) We only test the binary bias.** We only test the bias in pairs via SEAT and WinoBias, for example, 'male'/'female' or 'young'/'old'. However, it is widely recognized that terms in gender, race, etc. can be multi-polar.

**2) Our result is limited to English, and both corpora and wordlist tend towards North American social biases.** The whole of our experiment is conducted in English. In addition, Brown and SST Corpora are collected entirely in the North American environment. So are the wordlists. Therefore, it is expected that they skew towards North American social biases. When such models are debiased under the North American environment, it is necessary to understand how effective they are when transferred to other cultures.

**3) The generalization of conceptor-aided debiasing techniques can be tested more exhaustively.** This work has tested it on gender and race, but it can also be tested on other types of bias such as religious bias and hate speech.

## Ethical Considerations

The definition and recognition of bias are subtle. For example, we have used simple traditional binary definitions of male and female to examine gender bias. This, of course, ignores a much wider variety of gender identities, thus introducing an implicit bias to the analysis. Similarly, studies on racial bias rely on possibly problematic definitions of race. Core to our debiasing method is the selection of the wordlists. Each wordlist carries its own implicit definitions of gender, race, and other important dimensions. Care should be used to ensure that they represent the desired categories. To this end, it is often useful to involve people from the communities whose language is being debiased to better represent their values and belief systems.

One should also be careful in the use of debiasing. Removing signals about race or gender is often beneficial in reducing discrimination or in producing better translations. It may also remove key features of models needed for analyses. For example, removing gender or race 'signal' from the model may severely hamper the use of that model in gender studies or work on critical race theory. "White-washing" models are not always a benefit; sometimes one wants to see the bias inherent in a corpus.

## Acknowledgements

First, we would like to thank the reviewers for their fruitful discussion with us. We would also like to thank Claire Daniele for her editorial support. Last but not least, we appreciate PennNLP members for their helpful comments.

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

## A Attribute Wordlist

The examples and sources of the attribute wordlists are given below. Due to space limitations, we would only provide up to 50 words for each list.

### A.1 Pronouns Wordlist

Words (in total 22): *son, mother, daughter, him, brother, girl, uncle, hers, grandfather, his, boy, her, father, she, sister, man, female, aunt, woman, grandmother, he, male*.

They are the concatenation of W7_terms and W8_terms from WEAT list[4].

### A.2 Extended Wordlist

Words (randomly 50 of 388): *paramour, abbesses, headmistress, stepson, gods, congressman, gents, uncle, hers, wizard, cowgirls, fiancees, adultress, sororal, ladies, sons, uncles, actors, beards, heiress, fellas, salesman, princess, empress, masters, chairwomen, miss, horsewomen, actor, mr., strongwoman, barons, andrology, busboy, prince, hens, womb, masseuse, lady, testosterone, daughter, girl, stateswoman, businessmen, women, fraternities, aunts, boys, abbot, heroine, . . .*

They are the concatenation of lists: WinoBias extra gendered words[5], GN-GloVe male's name[6], and female's name[7].

### A.3 Propernouns Wordlist

Words (randomly 50 of 7578): *Broddie, Tony, Tawsha, Emylee, Orelle, Gerrilee, Katusha, Georges, Reine, Hayley, Deloria, Richmond, Wilfrid, Neille, Florie, Riva, Sandro, Cooper, Thom, Pate, Nikoletta, Rodrique, Pat, Chuck, Theressa, Brett, Kaspar, Elric, Storm, Yule, Bubba, Thomasina, Anson, Margery, Abra, Benedict, Cy, Gertrud, Morly, Julina, Melly, Quinta, Paolo, Brynne, Maurene, Alexis, Ramsey, Sianna, Phebe, Alfred, . . .*

They are the concatenation of lists: CMU male's name[8] and female's name[9].

---

[4] https://github.com/jsedoc/ConceptorDebias/blob/master/lists/WEAT_lists.py

[5] https://github.com/uclanlp/corefBias/blob/master/_site/WinoBias/wino/extra_gendered_words.txt

[6] https://github.com/uclanlp/gn_glove/blob/master/wordlist/female_word_file.txt

[7] https://github.com/uclanlp/gn_glove/blob/master/wordlist/male_word_file.txt

[8] https://www.cs.cmu.edu/Groups/AI/areas/nlp/corpora/names/male.txt

[9] https://www.cs.cmu.edu/Groups/AI/areas/nlp/corpora/names/female.txt

### A.4 Race Wordlist

Words (in total 8): *africa, african, america, asia, asian, caucasian, china, europe*

### A.5 Outlier Filtering

Table 6 shows the number of remaining gender words per percentile after being filtered on UMAP-clustering space.

| Percentile | Bias Subspace Type | | | |
|---|---|---|---|---|
| | pronouns | extended | propernouns | all |
| 0.5-1.0 | 22 | 388 | 7578 | 7988 |
| 0.4 | 22 | 388 | 5443 | 6902 |
| 0.3 | 11 | 372 | 4942 | 5140 |
| 0.2 | 7 | 364 | 3194 | 3289 |
| 0.1 | 4 | 67 | 1067 | 1087 |

Table 6: The number of remaining words per percentile after filtering on UMAP-clustering space. The model is "bert-base-uncased".

## B Model Checkpoints

We use the Hugging Face Transformers package (Wolf et al., 2019) in our experiments. The models and checkpoint names are given in Table 7.

| Model | Checkpoint |
|---|---|
| BERT-T | prajjwal1/bert-tiny |
| BERT | bert-base-uncased |
| BERT-L | bert-large-uncased |
| GPT2 | gpt2 |
| GPT2-L | gpt2-large |
| GPT-J | EleutherAI/gpt-j-6B |

Table 7: The package's model and checkpoint name in our experiment.

## C Continued Training Details

The conceptor-intervened model is trained for one epoch by setting prediction_loss_only as true and per_device_train_batch_size as 8. Following the training procedure in Devlin et al. (2019), we train by tasks Masked Language Model (MLM) and Next Sentence Prediction (NSP) simultaneously. The training corpus is the Wikipedia dump from datasets library (Lhoest et al., 2021).

## D GLUE Details

Before being evaluated on GLUE, each model is trained for three epochs with the following settings: batch_size 32, maximum_sequence_length 128, and learning_rate 2e−5; the same as Meade et al. (2022).

# E Full Bert-Base-Uncased Model Results

- Table 8 shows the gender debiasing result by different types of the corpus, using the last layer of "bert-base-uncased" as a benchmark.

- Table 9, and 10, 11 show the post-processing gender debiasing result of different percentiles of wordlist on three different corpora: Brown, SST, and Reddit, respectively.

- Table 12 and 13 show the post-processing and conceptor-intervened gender debiasing result of each layer on two different corpora: Brown and SST, respectively.

- Table 14 contains GLUE results for the gender debiased model.

# F Full Bert-Tiny Model Results

- Table 15, 16, and 17 show the post-processing gender debiasing result of different percentiles of wordlist on three different corpora: Brown, SST, and Reddit, respectively.

- Table 18 shows the post-processing and conceptor-intervened gender debiasing result of each layer on the SST corpus.

# G Full GPT2 Model Debiasing Results

- Table 19 shows the post-processing gender debiasing result of different percentiles of wordlist on Brown corpus.

# H Full Other LLMs' GLUE Results

- Table 20 contains GLUE results for gender debiased model.

# I Full Intersectional Debiasing Results

- Table 21 and 22 show the post-processing intersectional debiasing results.

| Model | SEAT-6 | SEAT-6b | SEAT-7 | SEAT-7b | SEAT-8 | SEAT-8b | Avg. Abs. | |
|---|---|---|---|---|---|---|---|---|
| BERT ("bert-base-uncased") | 0.931* | 0.090 | -0.124 | 0.937* | 0.783* | 0.858* | | 0.620 |
| *(Brown Corpus)* | | | | | | | | |
| + Conceptor-12 (pronouns) | **0.488*** | -0.091 | -0.331 | **0.471*** | 0.783* | **0.621*** | ↓0.156 | **0.464** |
| + Conceptor-12 (extended) | **0.509*** | -0.109 | -0.406 | **0.240** | **0.606*** | **0.449*** | ↓0.234 | **0.386** |
| + Conceptor-12 (propernouns) | **0.581*** | **-0.053** | -0.258 | **0.187** | **0.585*** | **0.659*** | ↓0.233 | **0.387** |
| + Conceptor-12 (all) | **0.452*** | -0.123 | -0.277 | **0.258** | **0.662*** | **0.607*** | ↓0.224 | **0.396** |
| + Conceptor-12 (or) | **0.440*** | **-0.063** | -0.136 | **0.251** | **0.640*** | **0.617*** | ↓0.262 | **0.358** |
| *(SST Corpus)* | | | | | | | | |
| + Conceptor-12 (pronouns) | **0.627*** | -0.104 | -0.416 | **0.520*** | **0.636*** | **0.628*** | ↓0.132 | **0.488** |
| + Conceptor-12 (extended) | **0.626*** | **-0.068** | -0.365 | **0.280** | **0.556*** | **0.429** | ↓0.233 | **0.387** |
| + Conceptor-12 (propernouns) | **0.680*** | **-0.050** | -0.405 | **0.614*** | **0.585*** | **0.790*** | ↓0.099 | **0.521** |
| + Conceptor-12 (all) | **0.624*** | -0.093 | -0.480 | **0.442*** | **0.538*** | **0.663*** | ↓0.147 | **0.473** |
| + Conceptor-12 (or) | **0.606*** | -0.100 | -0.447 | **0.337** | **0.428** | **0.575*** | ↓0.204 | **0.416** |
| *(Reddit Corpus)* | | | | | | | | |
| + Conceptor-12 (pronouns) | **0.619*** | -0.092 | -0.235 | **0.816*** | **0.756*** | 0.962* | ↓0.040 | **0.580** |
| + Conceptor-12 (extended) | **0.630*** | **-0.061** | -0.157 | **0.676*** | **0.711*** | **0.806*** | ↓0.113 | **0.507** |
| + Conceptor-12 (propernouns) | **0.792*** | 0.151 | **0.068** | 0.964* | **0.765*** | 0.934* | ↓0.008 | **0.612** |
| + Conceptor-12 (all) | **0.613*** | **-0.010** | **0.004** | **0.803*** | **0.735*** | 0.917* | ↓0.106 | **0.514** |
| + Conceptor-12 (or) | **0.593*** | 0.004 | **0.092** | **0.838*** | **0.652*** | 0.961* | ↓0.097 | **0.523** |
| + CDA | **0.846*** | 0.186 | -0.278 | 1.342* | **0.831*** | **0.849*** | ↑0.120 | 0.722 |
| + DROPOUT | **1.136*** | 0.317 | 0.138 | 1.179* | **0.879*** | **0.939*** | ↑0.144 | 0.765 |
| + INLP | **0.317** | -0.354 | -0.258 | **0.105** | **0.187** | **-0.004** | ↓0.416 | 0.204 |
| + SENTENCEDEBIAS | **0.350** | -0.298 | -0.626 | **0.458*** | **0.413** | **0.462*** | ↓0.186 | **0.434** |

Table 8: SEAT effect size of gender debising. The impact of *different corpora* on *bert-base-uncased* models. Effect sizes closer to 0 are indicative of less biased sentence representations (**bolded value**). Statistically significant effect sizes at $p < 0.01$ are denoted by *. Note that the "conceptor-$X$ (subspace)" indicates the conceptor negation matrix is generated by the $X$-layer of the language model in combinations with the subspace of the specific attribute wordlist. The top-3 best performance is colored in orange.

| Model | SEAT-6 | SEAT-6b | SEAT-7 | SEAT-7b | SEAT-8 | SEAT-8b | | Avg. Abs. |
|---|---|---|---|---|---|---|---|---|
| BERT ("bert-base-uncased") | 0.931* | 0.090 | -0.124 | 0.937* | 0.783* | 0.858* | | 0.620 |
| *(Wordlist Percentile 0.5-1.0)* | | | | | | | | |
| + Conceptor-12 (pronouns) | **0.488**\* | -0.091 | -0.331 | **0.471**\* | 0.783* | **0.621**\* | ↓0.156 | **0.464** |
| + Conceptor-12 (extended) | **0.509**\* | -0.109 | -0.406 | **0.240** | 0.606* | 0.449* | ↓0.234 | **0.386** |
| + Conceptor-12 (propernouns) | **0.581**\* | **-0.053** | -0.258 | **0.187** | 0.585* | 0.659* | ↓0.233 | **0.387** |
| + Conceptor-12 (all) | **0.452**\* | -0.123 | -0.277 | **0.258** | 0.662* | 0.607* | ↓0.224 | **0.396** |
| + Conceptor-12 (or) | **0.440**\* | **-0.063** | -0.136 | **0.251** | 0.640* | 0.617* | ↓0.262 | **0.358** |
| *(Wordlist Percentile 0.4)* | | | | | | | | |
| + Conceptor-12 (pronouns) | **0.483**\* | -0.095 | -0.385 | **0.435**\* | 0.776* | 0.609* | ↓0.156 | **0.464** |
| + Conceptor-12 (extended) | **0.509**\* | -0.110 | -0.407 | **0.239** | 0.603* | 0.447* | ↓0.234 | **0.386** |
| + Conceptor-12 (propernouns) | **0.451**\* | -0.188 | -0.505 | **-0.122** | 0.399 | 0.264 | ↓0.298 | **0.322** |
| + Conceptor-12 (all) | **0.466**\* | -0.112 | -0.260 | **0.267** | 0.697* | 0.617* | ↓0.217 | **0.403** |
| + Conceptor-12 (or) | **0.388** | **-0.078** | -0.292 | **0.179** | 0.594* | 0.335 | ↓0.309 | **0.311** |
| *(Wordlist Percentile 0.3)* | | | | | | | | |
| + Conceptor-12 (pronouns) | **0.487**\* | **-0.016** | -0.351 | **0.398**\* | 0.807* | **0.776**\* | ↓0.148 | **0.472** |
| + Conceptor-12 (extended) | **0.509**\* | -0.109 | -0.410 | **0.228** | 0.604* | 0.453* | ↓0.235 | **0.385** |
| + Conceptor-12 (propernouns) | **0.495**\* | -0.168 | -0.478 | **-0.083** | 0.456* | 0.315 | ↓0.288 | **0.332** |
| + Conceptor-12 (all) | **0.348** | -0.236 | -0.520 | **-0.019** | 0.506* | 0.361 | ↓0.288 | **0.332** |
| + Conceptor-12 (or) | **0.407** | **-0.022** | -0.247 | **0.331** | 0.677* | 0.483* | ↓0.259 | **0.361** |
| *(Wordlist Percentile 0.2)* | | | | | | | | |
| + Conceptor-12 (pronouns) | **0.570**\* | **0.035** | -0.378 | **0.334** | 0.708* | 0.768* | ↓0.154 | **0.466** |
| + Conceptor-12 (extended) | **0.508**\* | -0.109 | -0.416 | **0.219** | 0.602* | 0.450* | ↓0.236 | **0.384** |
| + Conceptor-12 (propernouns) | **0.548**\* | -0.157 | -0.397 | **0.270** | 0.483* | 0.366 | ↓0.250 | **0.370** |
| + Conceptor-12 (all) | **0.357** | -0.235 | -0.598 | **0.110** | 0.455* | 0.383 | ↓0.264 | **0.356** |
| + Conceptor-12 (or) | **0.476**\* | **-0.063** | -0.385 | **0.296** | 0.558* | 0.500* | ↓0.240 | **0.380** |
| *(Wordlist Percentile 0.1)* | | | | | | | | |
| + Conceptor-12 (pronouns) | **0.800**\* | 0.204 | -0.314 | **0.273** | 0.764* | 0.965* | ↓0.067 | **0.553** |
| + Conceptor-12 (extended) | **0.869**\* | 0.162 | -0.265 | **0.266** | 0.861* | 0.635* | ↓0.110 | **0.510** |
| + Conceptor-12 (propernouns) | **0.613**\* | **-0.084** | -0.582 | **0.190** | 0.579* | 0.740* | ↓0.155 | **0.465** |
| + Conceptor-12 (all) | **0.603**\* | -0.102 | -0.612 | **0.182** | 0.566* | 0.712* | ↓0.157 | **0.463** |
| + Conceptor-12 (or) | **0.614**\* | 0.197 | -0.401 | **-0.132** | 0.624* | 0.699* | ↓0.176 | **0.444** |
| + CDA | **0.846**\* | 0.186 | -0.278 | 1.342* | 0.831* | **0.849**\* | ↑0.120 | 0.722 |
| + DROPOUT | **1.136**\* | 0.317 | 0.138 | 1.179* | 0.879* | 0.939* | ↑0.144 | 0.765 |
| + INLP | **0.317** | -0.354 | -0.258 | **0.105** | 0.187 | -0.004 | ↓0.416 | **0.204** |
| + SENTENCEDEBIAS | **0.350** | -0.298 | -0.626 | **0.458**\* | 0.413 | 0.462* | ↓0.186 | **0.434** |

Table 9: SEAT effect size of gender debising. The impact of *different percentiles of wordlist* (using *UMAP* clustering) on *Brown* Corpus, *bert-base-uncased* models. The top-3 best performance is colored in orange.

| Model | SEAT-6 | SEAT-6b | SEAT-7 | SEAT-7b | SEAT-8 | SEAT-8b | Avg. Abs. | |
|---|---|---|---|---|---|---|---|---|
| BERT ("bert-base-uncased") | 0.931* | 0.090 | -0.124 | 0.937* | 0.783* | 0.858* | | 0.620 |
| *(Wordlist Percentile 0.5-1.0)* | | | | | | | | |
| + Conceptor-12 (pronouns) | **0.627*** | -0.104 | -0.416 | **0.520*** | **0.636*** | **0.628*** | ↓0.132 | **0.488** |
| + Conceptor-12 (extended) | **0.688*** | **0.024** | -0.293 | **0.138** | **0.559*** | 0.375 | ↓0.274 | **0.346** |
| + Conceptor-12 (propernouns) | **0.680*** | **-0.050** | -0.405 | **0.614*** | **0.585*** | **0.790*** | ↓0.099 | **0.521** |
| + Conceptor-12 (all) | **0.624*** | -0.093 | -0.480 | **0.442*** | **0.538*** | **0.663*** | ↓0.147 | **0.473** |
| + Conceptor-12 (or) | **0.619*** | **-0.069** | -0.428 | 0.280 | 0.414 | 0.539* | ↓0.229 | **0.391** |
| *(Wordlist Percentile 0.4)* | | | | | | | | |
| + Conceptor-12 (pronouns) | **0.619*** | -0.113 | -0.526 | **0.449*** | **0.606*** | **0.584*** | ↓0.137 | **0.483** |
| + Conceptor-12 (extended) | **0.688*** | **0.024** | -0.293 | **0.138** | **0.559*** | 0.375 | ↓0.274 | **0.346** |
| + Conceptor-12 (propernouns) | **0.704*** | **-0.086** | -0.227 | **0.590*** | **0.682*** | **0.716*** | ↓0.119 | **0.501** |
| + Conceptor-12 (all) | **0.622*** | **-0.087** | -0.508 | 0.277 | **0.519*** | **0.578*** | ↓0.188 | **0.432** |
| + Conceptor-12 (or) | **0.646*** | **-0.034** | -0.427 | 0.200 | **0.438*** | 0.401 | ↓0.262 | **0.358** |
| *(Wordlist Percentile 0.3)* | | | | | | | | |
| + Conceptor-12 (pronouns) | **0.550*** | **-0.035** | -0.396 | 0.344 | **0.682*** | **0.744*** | ↓0.161 | **0.459** |
| + Conceptor-12 (extended) | **0.687*** | **0.023** | -0.299 | **0.129** | **0.559*** | 0.382 | ↓0.273 | **0.347** |
| + Conceptor-12 (propernouns) | **0.706*** | **-0.088** | -0.230 | **0.602*** | **0.683*** | **0.720*** | ↓0.115 | **0.505** |
| + Conceptor-12 (all) | **0.652*** | -0.117 | -0.378 | **0.504*** | **0.536*** | **0.657*** | ↓0.146 | **0.474** |
| + Conceptor-12 (or) | **0.595*** | 0.027 | -0.375 | 0.171 | **0.519*** | **0.600*** | ↓0.239 | **0.381** |
| *(Wordlist Percentile 0.2)* | | | | | | | | |
| + Conceptor-12 (pronouns) | **0.730*** | 0.090 | **-0.110** | **0.523*** | **0.714*** | **0.758*** | ↓0.132 | **0.488** |
| + Conceptor-12 (extended) | **0.687*** | **0.023** | -0.299 | **0.129** | **0.559*** | 0.382 | ↓0.273 | **0.347** |
| + Conceptor-12 (propernouns) | **0.755*** | **-0.057** | -0.346 | **0.584*** | **0.659*** | **0.733*** | ↓0.098 | **0.522** |
| + Conceptor-12 (all) | **0.699*** | **-0.094** | -0.492 | **0.456*** | **0.620*** | **0.688*** | ↓0.112 | **0.508** |
| + Conceptor-12 (or) | **0.579*** | **-0.004** | -0.261 | 0.270 | **0.503*** | **0.634*** | ↓0.245 | **0.375** |
| *(Wordlist Percentile 0.1)* | | | | | | | | |
| + Conceptor-12 (pronouns) | **0.903*** | 0.198 | -0.464 | **0.030** | 0.434 | 0.492* | ↓0.200 | **0.420** |
| + Conceptor-12 (extended) | **0.631*** | -0.107 | **-0.012** | 1.008* | **0.615*** | **0.794*** | ↓0.092 | **0.528** |
| + Conceptor-12 (propernouns) | **0.655*** | -0.130 | -0.166 | **0.895*** | **0.641*** | 0.877* | ↓0.059 | **0.561** |
| + Conceptor-12 (all) | **0.597*** | -0.164 | -0.223 | **0.791*** | **0.698*** | 0.909* | ↓0.056 | **0.564** |
| + Conceptor-12 (or) | **0.542*** | **-0.039** | -0.184 | **0.589*** | **0.457*** | 0.929* | ↓0.163 | **0.457** |
| + CDA | **0.846*** | 0.186 | -0.278 | 1.342* | 0.831* | **0.849*** | ↑0.120 | 0.722 |
| + DROPOUT | **1.136*** | 0.317 | 0.138 | 1.179* | 0.879* | 0.939* | ↑0.144 | 0.765 |
| + INLP | 0.317 | -0.354 | -0.258 | **0.105** | 0.187 | **-0.004** | ↓0.416 | **0.204** |
| + SENTENCEDEBIAS | **0.350** | -0.298 | -0.626 | **0.458*** | 0.413 | 0.462* | ↓0.186 | **0.434** |

Table 10: SEAT effect size of gender debising. The impact of *different percentiles of wordlist* (using *UMAP* clustering) on *SST* Corpus, *bert-base-uncased* models. The top-3 best performance is colored in orange.

| Model | SEAT-6 | SEAT-6b | SEAT-7 | SEAT-7b | SEAT-8 | SEAT-8b | Avg. Abs. | |
|---|---|---|---|---|---|---|---|---|
| BERT ("bert-base-uncased") | 0.931* | 0.090 | -0.124 | 0.937* | 0.783* | 0.858* | | 0.620 |
| *(Wordlist Percentile 0.5-1.0)* | | | | | | | | |
| + Conceptor-12 (pronouns) | **0.619*** | -0.092 | -0.235 | **0.816*** | **0.756*** | 0.962* | ↓0.040 | **0.580** |
| + Conceptor-12 (extended) | **0.630*** | **-0.061** | -0.157 | **0.676*** | **0.711*** | **0.806*** | ↓0.113 | **0.507** |
| + Conceptor-12 (propernouns) | **0.792*** | 0.151 | **0.068** | 0.964* | **0.765*** | 0.934* | ↓0.008 | **0.612** |
| + Conceptor-12 (all) | **0.613*** | **-0.010** | **0.004** | **0.803*** | **0.735*** | 0.917* | ↓0.106 | **0.514** |
| + Conceptor-12 (or) | **0.593*** | **0.004** | **0.092** | **0.838*** | **0.652*** | 0.961* | ↓0.097 | **0.523** |
| *(Wordlist Percentile 0.4)* | | | | | | | | |
| + Conceptor-12 (pronouns) | **0.618*** | -0.095 | -0.297 | **0.769*** | **0.728*** | 0.944* | ↓0.045 | **0.575** |
| + Conceptor-12 (extended) | **0.732*** | **0.047** | **-0.121** | **0.601*** | **0.748*** | **0.719*** | ↓0.125 | **0.495** |
| + Conceptor-12 (propernouns) | **0.691*** | **-0.063** | 0.315 | 1.028* | **0.667*** | **0.635*** | ↓0.054 | **0.566** |
| + Conceptor-12 (all) | **0.651*** | **-0.010** | **-0.004** | **0.786*** | **0.732*** | 0.940* | ↓0.100 | **0.520** |
| + Conceptor-12 (or) | **0.526*** | **-0.057** | **0.054** | **0.596*** | **0.655*** | **0.724*** | ↓0.185 | **0.435** |
| *(Wordlist Percentile 0.3)* | | | | | | | | |
| + Conceptor-12 (pronouns) | **0.596*** | **-0.068** | -0.286 | **0.772*** | 0.790* | 0.983* | ↓0.038 | **0.582** |
| + Conceptor-12 (extended) | **0.740*** | **0.043** | **-0.119** | **0.593*** | 0.758* | **0.725*** | ↓0.124 | **0.496** |
| + Conceptor-12 (propernouns) | **0.825*** | **0.007** | 0.512* | 1.180* | 0.761* | **0.705*** | ↑0.045 | **0.665** |
| + Conceptor-12 (all) | **0.689*** | **0.007** | 0.180 | **0.924*** | **0.627*** | **0.653*** | ↓0.107 | **0.513** |
| + Conceptor-12 (or) | **0.612*** | **-0.018** | **0.046** | **0.745*** | **0.778*** | 0.916* | ↓0.101 | **0.519** |
| *(Wordlist Percentile 0.2)* | | | | | | | | |
| + Conceptor-12 (pronouns) | **0.801*** | 0.104 | **-0.072** | **0.757*** | 0.873* | 0.997* | ↓0.019 | **0.601** |
| + Conceptor-12 (extended) | **0.740*** | **0.043** | **-0.119** | **0.593*** | 0.758* | **0.725*** | ↓0.124 | **0.496** |
| + Conceptor-12 (propernouns) | **0.837*** | **0.037** | 0.532* | 1.170* | 0.785* | **0.722*** | ↑0.060 | **0.680** |
| + Conceptor-12 (all) | **0.694*** | **-0.042** | 0.356 | 1.044* | **0.608*** | **0.554*** | ↓0.070 | **0.550** |
| + Conceptor-12 (or) | **0.665*** | 0.096 | 0.130 | **0.607*** | 0.852* | 0.842* | ↓0.088 | **0.532** |
| *(Wordlist Percentile 0.1)* | | | | | | | | |
| + Conceptor-12 (pronouns) | 0.949* | 0.121 | -0.472 | **0.102** | 0.568* | 0.707* | ↓0.134 | **0.486** |
| + Conceptor-12 (extended) | **0.736*** | **-0.035** | -0.291 | **0.780*** | 0.812* | 1.095* | ↑0.005 | **0.625** |
| + Conceptor-12 (propernouns) | 0.948* | 0.107 | **-0.094** | 0.949* | 0.783* | **0.842*** | – | **0.620** |
| + Conceptor-12 (all) | 0.949* | 0.105 | **-0.061** | 0.899* | 0.782* | 0.846* | ↓0.013 | **0.607** |
| + Conceptor-12 (or) | 0.936* | 0.130 | -0.579 | 0.098 | 0.591* | 0.914* | ↓0.079 | **0.541** |
| + CDA | **0.846*** | 0.186 | -0.278 | 1.342* | 0.831* | **0.849*** | ↑0.120 | 0.722 |
| + DROPOUT | **1.136*** | 0.317 | 0.138 | 1.179* | 0.879* | 0.939* | ↑0.144 | 0.765 |
| + INLP | **0.317** | -0.354 | -0.258 | **0.105** | 0.187 | **-0.004** | ↓0.416 | **0.204** |
| + SENTENCEDEBIAS | **0.350** | -0.298 | -0.626 | **0.458*** | 0.413 | 0.462* | ↓0.186 | **0.434** |

Table 11: SEAT effect size of gender debising. The impact of *different percentiles of wordlist* (using *UMAP* clustering) on *Reddit* Corpus, *bert-base-uncased* models. The top-3 best performance is colored in orange.

| Model | SEAT-6 | SEAT-6b | SEAT-7 | SEAT-7b | SEAT-8 | SEAT-8b | | Avg. Abs. |
|---|---|---|---|---|---|---|---|---|
| (Layer 0) | | | | | | | | |
| BERT ("bert-base-uncased") | 0.921* | 0.194 | 0.251 | -0.172 | -0.110 | 0.366 | | 0.336 |
| + Conceptor-0 (or) | **0.147** | **-0.087** | -0.266 | -0.653 | -0.405 | **-0.324** | ↓0.022 | **0.314** |
| + Conceptor-Intervened | **0.147** | **-0.087** | -0.266 | -0.653 | -0.405 | **-0.324** | ↓0.022 | **0.314** |
| (Layer 1) | | | | | | | | |
| BERT ("bert-base-uncased") | 1.245* | 0.292 | 0.469* | 1.101* | 0.110 | 1.261* | | 0.746 |
| + Conceptor-1 (or) | **0.473*** | **0.205** | **-0.210** | **-0.093** | **-0.095** | 0.396 | ↓0.501 | **0.245** |
| + Conceptor-Intervened | **0.241** | **-0.038** | **-0.274** | 0.291 | -0.751 | **-0.107** | ↓0.462 | **0.284** |
| (Layer 2) | | | | | | | | |
| BERT ("bert-base-uncased") | 1.149* | 0.216 | 0.431* | 1.021* | 0.474* | 1.231* | | 0.754 |
| + Conceptor-2 (or) | **0.180** | **-0.047** | -0.450 | **-0.105** | 0.133 | 0.133 | ↓0.579 | **0.175** |
| + Conceptor-Intervened | **-0.108** | 0.115 | -0.965 | 1.388* | -1.146 | 0.329 | ↓0.079 | **0.675** |
| (Layer 3) | | | | | | | | |
| BERT ("bert-base-uncased") | 1.186* | 0.214 | 0.152 | 0.770* | 0.262 | 1.049* | | 0.606 |
| + Conceptor-3 (or) | **0.404** | **-0.046** | -0.675 | **0.102** | -0.325 | **-0.024** | ↓0.343 | **0.263** |
| + Conceptor-Intervened | **0.158** | 0.081 | -0.959 | 1.348* | -1.093 | 0.409 | ↑0.069 | **0.675** |
| (Layer 4) | | | | | | | | |
| BERT ("bert-base-uncased") | 0.975* | 0.106 | 0.552* | 0.890* | 0.542* | 0.724* | | 0.632 |
| + Conceptor-4 (or) | **0.597*** | **0.068** | **-0.249** | 0.251 | **-0.016** | **-0.315** | ↓0.383 | **0.249** |
| + Conceptor-Intervened | **0.060** | 0.121 | -0.986 | 1.676* | -0.840 | 0.943* | ↑0.139 | **0.771** |
| (Layer 5) | | | | | | | | |
| BERT ("bert-base-uncased") | 1.002* | 0.184 | 0.628* | 0.914* | 0.376 | 1.053* | | 0.693 |
| + Conceptor-5 (or) | **0.634*** | **0.064** | **0.118** | 0.225 | **-0.160** | 0.429 | ↓0.421 | **0.272** |
| + Conceptor-Intervened | **-0.046** | 0.043 | -1.038 | 1.378* | -0.790 | **0.659*** | ↓0.034 | **0.659** |
| (Layer 6) | | | | | | | | |
| BERT ("bert-base-uncased") | 0.753* | 0.118 | 0.539* | 1.048* | 0.597* | 1.042* | | 0.683 |
| + Conceptor-6 (or) | **0.327** | **0.041** | **0.176** | **0.104** | 0.150 | 0.174 | ↓0.521 | **0.162** |
| + Conceptor-Intervened | **-0.210** | 0.004 | -0.965 | 1.242* | -0.739 | **0.475*** | ↓0.077 | **0.606** |
| (Layer 7) | | | | | | | | |
| BERT ("bert-base-uncased") | 0.719* | 0.155 | 0.341 | 0.935* | 0.562* | 0.721* | | 0.572 |
| + Conceptor-7 (or) | **0.235** | **-0.064** | **-0.038** | **0.206** | 0.173 | 0.223 | ↓0.416 | **0.156** |
| + Conceptor-Intervened | **-0.246** | **-0.082** | -0.821 | 1.112* | -0.671 | 0.248 | ↓0.042 | **0.530** |
| (Layer 8) | | | | | | | | |
| BERT ("bert-base-uncased") | 0.983* | 0.163 | 0.313 | 1.157* | 0.766* | 0.789* | | 0.695 |
| + Conceptor-8 (or) | **0.235** | **0.005** | **-0.136** | 0.389 | 0.379 | 0.135 | ↓0.482 | **0.213** |
| + Conceptor-Intervened | **-0.125** | -0.193 | -0.940 | **0.796*** | -0.606 | 0.084 | ↓0.238 | **0.457** |
| (Layer 9) | | | | | | | | |
| BERT ("bert-base-uncased") | 0.922* | 0.224 | 0.503* | 1.293* | 0.780* | 0.996* | | 0.786 |
| + Conceptor-9 (or) | **0.234** | **0.019** | **-0.005** | **0.485*** | 0.694* | 0.686* | ↓0.432 | **0.354** |
| + Conceptor-Intervened | **-0.151** | -0.246 | -0.599 | **0.836*** | -0.455 | -0.095 | ↓0.389 | **0.397** |
| (Layer 10) | | | | | | | | |
| BERT ("bert-base-uncased") | 0.686* | 0.082 | 0.226 | 0.894* | 0.904* | 0.965* | | 0.626 |
| + Conceptor-10 (or) | **0.294** | -0.091 | **-0.153** | 0.078 | 0.703* | 0.545* | ↓0.315 | **0.311** |
| + Conceptor-Intervened | **-0.253** | -0.298 | -0.569 | **0.753*** | -0.462 | -0.099 | ↓0.221 | **0.405** |
| (Layer 11) | | | | | | | | |
| BERT ("bert-base-uncased") | 0.665* | -0.015 | -0.344 | 0.602* | 0.919* | 0.891* | | 0.573 |
| + Conceptor-11 (or) | **0.197** | -0.114 | -0.399 | **-0.157** | 0.557* | 0.277 | ↓0.289 | **0.284** |
| + Conceptor-Intervened | **-0.314** | -0.269 | -0.635 | 0.769* | -0.430 | 0.096 | ↓0.154 | **0.419** |
| (Layer 12) | | | | | | | | |
| BERT ("bert-base-uncased") | 0.931* | 0.090 | -0.124 | 0.937* | 0.783* | 0.858* | | 0.620 |
| + Conceptor-12 (or) | **0.388** | **-0.078** | -0.292 | 0.179 | 0.594* | 0.335 | ↓0.309 | **0.311** |
| + Conceptor-Intervened | **-0.334** | -0.117 | -0.698 | **0.459*** | -0.230 | 0.178 | ↓0.284 | **0.336** |
| + CDA | **0.846*** | 0.186 | -0.278 | 1.342* | 0.831* | **0.849*** | ↑0.120 | 0.722 |
| + Dropout | **1.136*** | 0.317 | 0.138 | 1.179* | 0.879* | 0.939* | ↑0.144 | 0.765 |
| + INLP | **0.317** | -0.354 | -0.258 | **0.105** | 0.187 | **-0.004** | ↓0.416 | **0.204** |
| + SentenceDebias | **0.350** | -0.298 | -0.626 | **0.458*** | 0.413 | 0.462* | ↓0.186 | **0.434** |

Table 12: SEAT fffect size of gender debising from CI-BERT, Type I. The conceptor-intervened performance of *different layer's conceptors* on *SST* Corpus, *bert-base-uncased* models. The setting is "brown-0.4-or". The layer(s) of CI-BERT that outperform the conceptor post-processing of the same layer(s) are colored in orange.

| Model | SEAT-6 | SEAT-6b | SEAT-7 | SEAT-7b | SEAT-8 | SEAT-8b | | Avg. Abs. |
|---|---|---|---|---|---|---|---|---|
| (Layer 0) | | | | | | | | |
| BERT ("bert-base-uncased") | 0.921* | 0.194 | 0.251 | -0.172 | -0.110 | 0.366 | | 0.336 |
| + Conceptor-0 (extended) | **0.497*** | **-0.095** | -0.412 | -0.760 | **-0.001** | -0.276 | ↑0.004 | **0.340** |
| + Conceptor-Intervened | **0.497*** | **-0.095** | -0.412 | -0.760 | **-0.001** | -0.276 | ↑0.004 | **0.340** |
| (Layer 1) | | | | | | | | |
| BERT ("bert-base-uncased") | 1.245* | 0.292 | 0.469* | 1.101* | 0.110 | 1.261* | | 0.746 |
| + Conceptor-1 (extended) | **0.897*** | **0.156** | **-0.084** | **0.208** | **0.099** | **0.558*** | ↓0.412 | **0.334** |
| + Conceptor-Intervened | **0.813*** | **0.029** | -0.961 | **-0.513** | -0.211 | -0.292 | ↓0.276 | **0.470** |
| (Layer 2) | | | | | | | | |
| BERT ("bert-base-uncased") | 1.149* | 0.216 | 0.431* | 1.021* | 0.474* | 1.231* | | 0.754 |
| + Conceptor-2 (extended) | **0.542*** | **0.059** | **-0.146** | **0.112** | **0.515*** | **0.428** | ↓0.454 | **0.300** |
| + Conceptor-Intervened | **0.366** | **0.088** | -1.342 | **-0.249** | -0.408 | -0.634 | ↓0.239 | **0.515** |
| (Layer 3) | | | | | | | | |
| BERT ("bert-base-uncased") | 1.186* | 0.214 | 0.152 | 0.770* | 0.262 | 1.049* | | 0.606 |
| + Conceptor-3 (extended) | **0.849*** | **-0.034** | -0.516 | **0.154** | **0.205** | **0.356** | ↓0.254 | **0.352** |
| + Conceptor-Intervened | **0.329** | **0.048** | -1.240 | **-0.520** | -0.429 | -0.499 | ↓0.095 | **0.511** |
| (Layer 4) | | | | | | | | |
| BERT ("bert-base-uncased") | 0.975* | 0.106 | 0.552* | 0.890* | 0.542* | 0.724* | | 0.632 |
| + Conceptor-4 (extended) | **0.789*** | 0.109 | **-0.014** | **0.254** | **0.515*** | **0.009** | ↓0.350 | **0.282** |
| + Conceptor-Intervened | **0.248** | **0.068** | -1.367 | **-0.040** | -0.401 | -0.100 | ↓0.261 | **0.371** |
| (Layer 5) | | | | | | | | |
| BERT ("bert-base-uncased") | 1.002* | 0.184 | 0.628* | 0.914* | 0.376 | 1.053* | | 0.693 |
| + Conceptor-5 (extended) | **0.695*** | **0.122** | **-0.007** | **0.075** | **0.158** | **0.472*** | ↓0.438 | **0.255** |
| + Conceptor-Intervened | **0.105** | **0.066** | -1.096 | **-0.170** | -0.173 | 0.036 | ↓0.419 | **0.274** |
| (Layer 6) | | | | | | | | |
| BERT ("bert-base-uncased") | 0.753* | 0.118 | 0.539* | 1.048* | 0.597* | 1.042* | | 0.683 |
| + Conceptor-6 (extended) | **0.372** | **0.084** | **0.150** | **0.033** | **0.467*** | **0.209** | ↓0.464 | **0.219** |
| + Conceptor-Intervened | **0.004** | **-0.023** | -0.866 | **-0.312** | -0.234 | -0.106 | ↓0.425 | **0.258** |
| (Layer 7) | | | | | | | | |
| BERT ("bert-base-uncased") | 0.719* | 0.155 | 0.341 | 0.935* | 0.562* | 0.721* | | 0.572 |
| + Conceptor-7 (extended) | **0.451*** | **0.082** | **0.051** | **0.196** | **0.326** | **0.185** | ↓0.357 | **0.215** |
| + Conceptor-Intervened | **0.041** | **-0.066** | -0.697 | **-0.509** | -0.381 | -0.015 | ↓0.287 | **0.285** |
| (Layer 8) | | | | | | | | |
| BERT ("bert-base-uncased") | 0.983* | 0.163 | 0.313 | 1.157* | 0.766* | 0.789* | | 0.695 |
| + Conceptor-8 (extended) | **0.597*** | **0.051** | **-0.023** | **0.639*** | **0.503*** | **0.200** | ↓0.359 | **0.336** |
| + Conceptor-Intervened | **0.110** | **-0.095** | -0.392 | **-0.702** | -0.287 | 0.190 | ↓0.399 | **0.296** |
| (Layer 9) | | | | | | | | |
| BERT ("bert-base-uncased") | 0.922* | 0.224 | 0.503* | 1.293* | 0.780* | 0.996* | | 0.786 |
| + Conceptor-9 (extended) | **0.597*** | **0.146** | **0.333** | **0.903*** | **0.764*** | **0.722*** | ↓0.208 | **0.578** |
| + Conceptor-Intervened | **0.148** | **-0.024** | **0.162** | -0.669 | 0.224 | 0.487* | ↓0.500 | **0.286** |
| (Layer 10) | | | | | | | | |
| BERT ("bert-base-uncased") | 0.686* | 0.082 | 0.226 | 0.894* | 0.904* | 0.965* | | 0.626 |
| + Conceptor-10 (extended) | **0.639*** | 0.099 | **-0.034** | 0.044 | **0.605*** | **0.322** | ↓0.335 | **0.291** |
| + Conceptor-Intervened | **0.557*** | -0.165 | **-0.149** | -1.046 | 0.142 | 0.522* | ↓0.196 | **0.430** |
| (Layer 11) | | | | | | | | |
| BERT ("bert-base-uncased") | 0.665* | -0.015 | -0.344 | 0.602* | 0.919* | 0.891* | | 0.573 |
| + Conceptor-11 (extended) | **0.565*** | **-0.045** | -0.511 | **-0.406** | **0.523*** | **0.198** | ↓0.198 | **0.375** |
| + Conceptor-Intervened | **0.602*** | **-0.189** | **0.143** | -1.219 | -0.006 | 0.205 | ↓0.179 | **0.394** |
| (Layer 12) | | | | | | | | |
| BERT ("bert-base-uncased") | 0.931* | 0.090 | -0.124 | 0.937* | 0.783* | 0.858* | | 0.620 |
| + Conceptor-12 (extended) | **0.688*** | **0.024** | -0.293 | **0.138** | **0.559*** | **0.375** | ↓0.274 | **0.346** |
| + Conceptor-Intervened | **0.384** | -0.261 | 0.144 | -1.256 | **-0.148** | 0.398 | ↓0.188 | **0.432** |
| + CDA | **0.846*** | 0.186 | -0.278 | 1.342* | 0.831* | **0.849*** | ↑0.120 | 0.722 |
| + Dropout | **1.136*** | 0.317 | 0.138 | 1.179* | 0.879* | 0.939* | ↑0.144 | 0.765 |
| + INLP | **0.317** | -0.354 | -0.258 | **0.105** | 0.187 | **-0.004** | ↓0.416 | **0.204** |
| + SentenceDebias | **0.350** | -0.298 | -0.626 | **0.458*** | 0.413 | 0.462* | ↓0.186 | **0.434** |

Table 13: SEAT effect size of gender debiasing from CI-BERT, Type I. The conceptor-intervened performance of *different layer's conceptors* on *SST* Corpus, *bert-base-uncased* models. The setting is "sst-0.9-extended". The layer(s) of CI-BERT that outperform the conceptor post-processing of the same layer(s) are colored in orange.

| Model | CoLA | MNLI | MRPC | QNLI | QQP | RTE | SST | STS-B | WNLI | | Average |
|---|---|---|---|---|---|---|---|---|---|---|---|
| BERT | 55.89 | 84.50 | 88.59 | 91.38 | 91.03 | 63.54 | 92.58 | 88.51 | 43.66 | | 77.74 |
| + Conceptor P.P | 57.54 | 84.66 | 89.30 | 91.03 | 91.05 | 65.34 | 92.66 | 89.07 | 54.93 | ↑1.77 | 79.51 |
| + Conceptor C.T. | 47.06 | 83.46 | 87.20 | 90.73 | 90.97 | 58.98 | 91.67 | 88.21 | 52.11 | ↓1.03 | 76.71 |
| + CDA | 55.90 | 84.73 | 88.76 | 91.36 | 91.01 | 66.31 | 92.43 | 89.14 | 38.03 | ↓0.22 | 77.52 |
| + Dropout | 49.83 | 84.67 | 88.20 | 91.27 | 90.36 | 64.02 | 92.58 | 88.47 | 37.09 | ↓1.46 | 76.28 |
| + INLP | 56.06 | 84.81 | 88.61 | 91.34 | 90.92 | 64.98 | 92.51 | 88.70 | 32.86 | ↓0.99 | 76.76 |
| + SentenceDebias | 56.41 | 84.80 | 88.70 | 91.48 | 90.98 | 63.06 | 92.32 | 88.45 | 44.13 | ↑0.07 | 77.81 |

Table 14: GLUE validation set results for gender debiased BERT model. We use the F1 score for MRPC, the Spearman correlation for STS-B, and Matthew's correlation for CoLA. For all other tasks, we report accuracy. All scores are averaged among three runs. The model is "bert-base-uncased". The top-3 best performance is colored in orange.

| Model | SEAT-6 | SEAT-6b | SEAT-7 | SEAT-7b | SEAT-8 | SEAT-8b | | Avg. Abs. |
|---|---|---|---|---|---|---|---|---|
| BERT-T | 1.735* | 0.797* | 1.294* | 1.243* | 0.837* | 1.293* | | 1.200 |
| (Wordlist Percentile 1.0) | | | | | | | | |
| + Conceptor-2 (pronouns) | **1.657*** | **0.449*** | **1.185*** | **0.936*** | **0.453*** | **0.833*** | ↓0.281 | **0.919** |
| + Conceptor-2 (extended) | **1.570*** | 0.353 | **1.094*** | **0.991*** | 0.176 | **0.775*** | ↓0.373 | **0.827** |
| + Conceptor-2 (propernouns) | **1.641*** | **0.655*** | **1.142*** | **1.121*** | 0.203 | **0.781*** | ↓0.276 | **0.924** |
| + Conceptor-2 (all) | **1.587*** | **0.377*** | **1.188*** | **1.077*** | 0.128 | **0.735*** | ↓0.351 | **0.849** |
| + Conceptor-2 (or) | **1.464*** | 0.257 | **1.005*** | **0.944*** | -0.114 | **0.503*** | ↓0.486 | 0.714 |
| (Wordlist Percentile 0.5-0.9) | | | | | | | | |
| + Conceptor-2 (pronouns) | **1.657*** | **0.449*** | **1.185*** | **0.936*** | **0.453*** | **0.833*** | ↓0.281 | **0.919** |
| + Conceptor-2 (extended) | **1.296*** | 0.255 | **1.014*** | **1.194*** | -0.274 | **0.502*** | ↓0.444 | **0.756** |
| + Conceptor-2 (propernouns) | **1.641*** | **0.655*** | **1.142*** | **1.121*** | 0.203 | **0.781*** | ↓0.276 | **0.924** |
| + Conceptor-2 (all) | **1.587*** | **0.377*** | **1.188*** | **1.077*** | 0.128 | **0.735*** | ↓0.351 | **0.849** |
| + Conceptor-2 (or) | **1.323*** | 0.186 | **0.947*** | **1.027*** | -0.289 | 0.403 | ↓0.504 | 0.696 |
| (Wordlist Percentile 0.4) | | | | | | | | |
| + Conceptor-2 (pronouns) | **1.657*** | **0.443*** | **1.181*** | **0.937*** | **0.448*** | **0.821*** | ↓0.286 | **0.914** |
| + Conceptor-2 (extended) | **1.294*** | 0.254 | **1.014*** | **1.194*** | -0.274 | **0.502*** | ↓0.445 | **0.755** |
| + Conceptor-2 (propernouns) | **1.589*** | **0.722*** | **1.130*** | **1.084*** | **0.494*** | **0.991*** | ↓0.198 | **1.002** |
| + Conceptor-2 (all) | **1.585*** | **0.396*** | **1.192*** | **1.067*** | 0.159 | **0.726*** | ↓0.346 | **0.854** |
| + Conceptor-2 (or) | **1.278*** | 0.233 | **0.852*** | **0.910*** | -0.265 | 0.346 | ↓0.553 | 0.647 |
| (Wordlist Percentile 0.3) | | | | | | | | |
| + Conceptor-2 (pronouns) | **1.691*** | **0.573*** | **1.227*** | **1.158*** | **0.573*** | **1.009*** | ↓0.161 | **1.039** |
| + Conceptor-2 (extended) | **1.295*** | 0.260 | **1.010*** | **1.192*** | -0.286 | **0.490*** | ↓0.444 | **0.756** |
| + Conceptor-2 (propernouns) | **1.597*** | **0.746*** | **1.162*** | **1.147*** | **0.551*** | **1.034*** | ↓0.160 | **1.040** |
| + Conceptor-2 (all) | **1.536*** | **0.436*** | **1.143*** | **1.181*** | 0.140 | **0.849*** | ↓0.319 | **0.881** |
| + Conceptor-2 (or) | **1.277*** | 0.235 | **1.055*** | **1.168*** | -0.090 | **0.542*** | ↓0.472 | **0.728** |
| (Wordlist Percentile 0.2) | | | | | | | | |
| + Conceptor-2 (pronouns) | **1.656*** | **0.543*** | **1.253*** | **1.175*** | **0.569*** | **1.038*** | ↓0.161 | **1.039** |
| + Conceptor-2 (extended) | **1.296*** | 0.260 | **1.011*** | **1.189*** | -0.290 | **0.478*** | ↓0.446 | **0.754** |
| + Conceptor-2 (propernouns) | **1.577*** | **0.723*** | **1.231*** | **1.193*** | **0.541*** | **1.067*** | ↓0.145 | **1.055** |
| + Conceptor-2 (all) | **1.490*** | 0.341 | **1.116*** | **1.182*** | -0.018 | **0.849*** | ↓0.367 | **0.833** |
| + Conceptor-2 (or) | **1.178*** | 0.141 | **1.062*** | **1.087*** | -0.252 | **0.502*** | ↓0.496 | **0.704** |
| (Wordlist Percentile 0.1) | | | | | | | | |
| + Conceptor-2 (pronouns) | **1.677*** | **0.643*** | 1.317* | 1.342* | **0.696*** | **1.191*** | ↓0.056 | **1.144** |
| + Conceptor-2 (extended) | **1.547*** | **0.700*** | 1.305* | 1.333* | **0.464*** | **0.997*** | ↓0.142 | **1.058** |
| + Conceptor-2 (propernouns) | **1.722*** | 0.836* | **1.256*** | **1.213*** | 0.956* | 1.316* | ↑0.017 | 1.217 |
| + Conceptor-2 (all) | 1.771* | 0.882* | **1.189*** | **1.160*** | 0.996* | **1.277*** | ↑0.012 | 1.212 |
| + Conceptor-2 (or) | **1.579*** | **0.560*** | **1.278*** | 1.301* | **0.422** | **0.881*** | ↓0.196 | **1.004** |

Table 15: SEAT effect size of gender debising. The impact of *different percentiles of wordlist* (using *UMAP* clustering) on *Brown* Corpus, *bert-tiny* models. The top-3 best performance is colored in orange.

| Model | SEAT-6 | SEAT-6b | SEAT-7 | SEAT-7b | SEAT-8 | SEAT-8b | Avg. Abs. | |
|---|---|---|---|---|---|---|---|---|
| BERT-T | 1.735* | 0.797* | 1.294* | 1.243* | 0.837* | 1.293* | | 1.200 |
| *(Wordlist Percentile 1.0)* | | | | | | | | |
| + Conceptor-2 (pronouns) | **1.703*** | **0.403*** | **0.958*** | **0.706*** | 0.254 | **0.679*** | ↓0.416 | **0.784** |
| + Conceptor-2 (extended) | **1.608*** | **0.473*** | **0.870*** | **1.118*** | -0.209 | **0.732*** | ↓0.365 | **0.835** |
| + Conceptor-2 (propernouns) | **1.704*** | **0.582*** | **1.012*** | **1.111*** | -0.069 | **0.730*** | ↓0.332 | **0.868** |
| + Conceptor-2 (all) | **1.680*** | **0.377*** | **1.028*** | **1.047*** | -0.175 | **0.669*** | ↓0.371 | **0.829** |
| + Conceptor-2 (or) | **1.489*** | 0.163 | **0.539*** | **0.937*** | -0.612 | 0.314 | ↓0.524 | **0.676** |
| *(Wordlist Percentile 0.5-0.9)* | | | | | | | | |
| + Conceptor-2 (pronouns) | **1.703*** | **0.403*** | **0.958*** | **0.706*** | 0.254 | **0.679*** | ↓0.416 | **0.784** |
| + Conceptor-2 (extended) | **1.647*** | **0.391*** | **0.806*** | **0.815*** | -0.136 | **0.637*** | ↓0.461 | **0.739** |
| + Conceptor-2 (propernouns) | **1.704*** | **0.582*** | **1.012*** | **1.111*** | -0.069 | **0.730*** | ↓0.332 | **0.868** |
| + Conceptor-2 (all) | **1.680*** | **0.377*** | **1.028*** | **1.047*** | -0.175 | **0.669*** | ↓0.371 | **0.829** |
| + Conceptor-2 (or) | **1.542*** | 0.148 | **0.486*** | **0.806*** | -0.549 | 0.245 | ↓0.571 | **0.629** |
| *(Wordlist Percentile 0.4)* | | | | | | | | |
| + Conceptor-2 (pronouns) | **1.703*** | **0.375*** | **0.999*** | **0.720*** | 0.235 | **0.669*** | ↓0.416 | **0.784** |
| + Conceptor-2 (extended) | **1.608*** | **0.473*** | **0.870*** | **1.118*** | -0.209 | **0.732*** | ↓0.365 | **0.835** |
| + Conceptor-2 (propernouns) | 1.765* | 0.954* | **1.027*** | **1.036*** | 0.457* | 1.028* | ↓0.156 | 1.044 |
| + Conceptor-2 (all) | **1.694*** | **0.444*** | **1.057*** | **1.054*** | -0.090 | **0.709*** | ↓0.359 | **0.841** |
| + Conceptor-2 (or) | **1.587*** | 0.344 | **0.512*** | **0.881*** | -0.407 | **0.489*** | ↓0.497 | **0.703** |
| *(Wordlist Percentile 0.3)* | | | | | | | | |
| + Conceptor-2 (pronouns) | 1.786* | 0.843* | **1.121*** | **1.028*** | 0.617* | 1.064* | ↓0.124 | 1.076 |
| + Conceptor-2 (extended) | **1.610*** | **0.479*** | **0.866*** | **1.119*** | -0.215 | **0.727*** | ↓0.364 | **0.836** |
| + Conceptor-2 (propernouns) | 1.749* | 0.945* | **1.039*** | **1.067*** | 0.474* | 1.041* | ↓0.148 | 1.052 |
| + Conceptor-2 (all) | 1.751* | 0.813* | **1.102*** | **1.063*** | 0.522* | 1.092* | ↓0.143 | 1.057 |
| + Conceptor-2 (or) | **1.652*** | **0.414*** | **0.784*** | **1.047*** | -0.170 | **0.725*** | ↓0.401 | **0.799** |
| *(Wordlist Percentile 0.2)* | | | | | | | | |
| + Conceptor-2 (pronouns) | 1.862* | 0.983* | **1.178*** | **0.982*** | 0.737* | 1.080* | ↓0.063 | 1.137 |
| + Conceptor-2 (extended) | **1.610*** | **0.479*** | **0.866*** | **1.119*** | -0.215 | **0.727*** | ↓0.364 | **0.836** |
| + Conceptor-2 (propernouns) | 1.751* | 0.893* | **1.082*** | **1.125*** | 0.581* | 1.167* | ↓0.100 | 1.100 |
| + Conceptor-2 (all) | 1.773* | 0.862* | **1.101*** | **1.120*** | 0.628* | 1.191* | ↓0.088 | 1.112 |
| + Conceptor-2 (or) | 1.736* | **0.582*** | **0.702*** | **0.946*** | -0.103 | **0.763*** | ↓0.395 | **0.805** |
| *(Wordlist Percentile 0.1)* | | | | | | | | |
| + Conceptor-2 (pronouns) | 1.828* | 0.971* | **1.185*** | **1.065*** | 0.755* | 1.123* | ↓0.046 | 1.154 |
| + Conceptor-2 (extended) | **1.638*** | **0.511*** | **1.167*** | **1.114*** | 0.265 | 1.017* | ↓0.248 | 0.952 |
| + Conceptor-2 (propernouns) | 1.777* | 0.941* | **1.121*** | **1.169*** | 0.885* | 1.332* | ↑0.004 | 1.204 |
| + Conceptor-2 (all) | 1.795* | 0.952* | **1.070*** | **1.129*** | 0.785* | 1.269* | ↓0.033 | 1.167 |
| + Conceptor-2 (or) | **1.706*** | **0.726*** | **0.990*** | **0.972*** | 0.455* | **0.978*** | ↓0.229 | **0.971** |

Table 16: SEAT effect size of gender debising. The impact of *different percentiles of wordlist* (using *UMAP* clustering) on *SST* Corpus, *bert-tiny* models. The top-3 best performance is colored in orange.

| Model | SEAT-6 | SEAT-6b | SEAT-7 | SEAT-7b | SEAT-8 | SEAT-8b | Avg. Abs. | |
|---|---|---|---|---|---|---|---|---|
| BERT-T | 1.735* | 0.797* | 1.294* | 1.243* | 0.837* | 1.293* | | 1.200 |
| *(Wordlist Percentile 1.0)* | | | | | | | | |
| + Conceptor-2 (pronouns) | **1.676*** | **0.389*** | **1.218*** | **1.095*** | **0.557*** | **1.008*** | ↓0.210 | **0.990** |
| + Conceptor-2 (extended) | **1.578*** | **0.507*** | **1.248*** | **1.220*** | **0.656*** | 1.351* | ↓0.107 | **1.093** |
| + Conceptor-2 (propernouns) | **1.713*** | **0.776*** | **1.184*** | 1.315* | **0.538*** | **1.193*** | ↓0.080 | **1.120** |
| + Conceptor-2 (all) | **1.660*** | **0.379*** | **1.248*** | **1.185*** | **0.486*** | **1.125*** | ↓0.186 | **1.014** |
| + Conceptor-2 (or) | **1.550*** | **0.180** | **1.010*** | **1.146*** | **0.197** | **1.088*** | ↓0.338 | **0.862** |
| *(Wordlist Percentile 0.5-0.9)* | | | | | | | | |
| + Conceptor-2 (pronouns) | **1.676*** | **0.389*** | **1.218*** | **1.095*** | **0.557*** | **1.008*** | ↓0.210 | **0.990** |
| + Conceptor-2 (extended) | **1.684*** | **0.374** | **1.204*** | **1.065*** | **0.616*** | **1.144*** | ↓0.186 | **1.014** |
| + Conceptor-2 (propernouns) | **1.713*** | **0.776*** | **1.184*** | 1.315* | **0.538*** | **1.193*** | ↓0.080 | **1.120** |
| + Conceptor-2 (all) | **1.660*** | **0.379*** | **1.248*** | **1.185*** | **0.486*** | **1.125*** | ↓0.186 | **1.014** |
| + Conceptor-2 (or) | **1.573*** | **0.179** | **0.963*** | **1.117*** | **0.103** | **0.963*** | ↓0.384 | **0.816** |
| *(Wordlist Percentile 0.4)* | | | | | | | | |
| + Conceptor-2 (pronouns) | **1.677*** | **0.382*** | **1.218*** | **1.095*** | **0.561*** | **1.008*** | ↓0.210 | **0.990** |
| + Conceptor-2 (extended) | **1.578*** | **0.507*** | **1.248*** | **1.220*** | **0.656*** | 1.351* | ↓0.107 | **1.093** |
| + Conceptor-2 (propernouns) | **1.622*** | **0.745*** | **0.946*** | **0.937*** | **0.717*** | **1.104*** | ↓0.188 | **1.012** |
| + Conceptor-2 (all) | **1.656*** | **0.388*** | **1.198*** | **1.114*** | **0.490*** | **1.083*** | ↓0.212 | **0.988** |
| + Conceptor-2 (or) | **1.544*** | **0.270** | **0.899*** | **1.034*** | **0.497*** | **1.063*** | ↓0.316 | **0.884** |
| *(Wordlist Percentile 0.3)* | | | | | | | | |
| + Conceptor-2 (pronouns) | **1.554*** | **0.447*** | **1.116*** | **1.198*** | 0.973* | 1.429* | ↓0.080 | **1.120** |
| + Conceptor-2 (extended) | **1.584*** | **0.523*** | **1.248*** | **1.223*** | **0.636*** | 1.345* | ↓0.107 | **1.093** |
| + Conceptor-2 (propernouns) | **1.648*** | **0.764*** | **1.087*** | **1.134*** | 1.084* | 1.349* | ↓0.022 | **1.178** |
| + Conceptor-2 (all) | **1.588*** | **0.783*** | **1.125*** | **1.132*** | 0.989* | 1.325* | ↓0.043 | **1.157** |
| + Conceptor-2 (or) | **1.549*** | **0.402*** | **1.236*** | **1.186*** | 1.024* | 1.430* | ↓0.062 | **1.138** |
| *(Wordlist Percentile 0.2)* | | | | | | | | |
| + Conceptor-2 (pronouns) | **1.653*** | **0.588*** | **1.124*** | **1.084*** | 0.863* | **1.257*** | ↓0.105 | **1.095** |
| + Conceptor-2 (extended) | **1.584*** | **0.523*** | **1.248*** | **1.223*** | **0.636*** | 1.345* | ↓0.107 | **1.093** |
| + Conceptor-2 (propernouns) | **1.623*** | **0.753*** | **1.184*** | **1.224*** | 1.001* | 1.336* | ↓0.013 | **1.187** |
| + Conceptor-2 (all) | **1.600*** | **0.747*** | **1.190*** | **1.195*** | 0.986* | 1.314* | ↓0.028 | **1.172** |
| + Conceptor-2 (or) | **1.476*** | **0.364** | **1.089*** | **1.026*** | 0.407 | **1.162*** | ↓0.279 | **0.921** |
| *(Wordlist Percentile 0.1)* | | | | | | | | |
| + Conceptor-2 (pronouns) | **1.668*** | **0.775*** | **1.154*** | **0.989*** | **0.813*** | **1.171*** | ↓0.105 | **1.095** |
| + Conceptor-2 (extended) | **1.689*** | **0.762*** | 1.345* | **1.206*** | 0.991* | **1.260*** | ↑0.009 | 1.209 |
| + Conceptor-2 (propernouns) | **1.705*** | 0.863* | **1.257*** | **1.214*** | 0.812* | **1.282*** | ↓0.011 | **1.189** |
| + Conceptor-2 (all) | **1.709*** | 0.870* | **1.224*** | **1.203*** | 0.858* | **1.287*** | ↓0.008 | **1.192** |
| + Conceptor-2 (or) | **1.630*** | **0.739*** | **1.069*** | **0.928*** | **0.753*** | **1.152*** | ↓0.155 | **1.045** |

Table 17: SEAT effect size of gender debising. The impact of *different percentiles of wordlist* (using *UMAP* clustering) on *Reddit* Corpus, *bert-tiny* models. The top-3 best performance is colored in orange.

| Model | SEAT-6 | SEAT-6b | SEAT-7 | SEAT-7b | SEAT-8 | SEAT-8b | Avg. Abs. | |
|---|---|---|---|---|---|---|---|---|
| *(Layer 0)* | | | | | | | | |
| BERT-T | 1.536* | 0.640* | 0.959* | 1.307* | 0.263 | 0.814* | | 0.920 |
| + Conceptor-0 (or) | **0.803*** | **0.103** | **0.249** | **0.825*** | **0.039** | **0.568*** | ↓0.489 | **0.431** |
| + Conceptor-Intervened | **0.803*** | **0.103** | **0.249** | **0.825*** | **0.039** | **0.568*** | ↓0.489 | **0.431** |
| *(Layer 1)* | | | | | | | | |
| BERT-T | 1.702* | 1.019* | 1.102* | 1.250* | 0.313 | 1.094* | | 1.080 |
| + Conceptor-1 (or) | **1.241*** | **-0.067** | **0.588*** | **0.939*** | **-0.340** | **0.477*** | ↓0.471 | **0.609** |
| + Conceptor-Intervened | **0.928*** | **0.022** | **-0.427** | **0.708*** | **-0.753** | **0.542*** | ↓0.517 | **0.563** |
| *(Layer 2)* | | | | | | | | |
| BERT-T | 1.735* | 0.797* | 1.294* | 1.243* | 0.837* | 1.293* | | 1.200 |
| + Conceptor-2 (or) | **1.542*** | **0.148** | **0.486*** | **0.806*** | **-0.549** | **0.245** | ↓0.571 | **0.629** |
| + Conceptor-Intervened | **1.026*** | **-0.079** | **-0.264** | **0.862*** | **-0.500** | **0.239** | ↓0.705 | **0.495** |

Table 18: SEAT effect size of gender debising from CI-BERT, Type I. The conceptor-intervened performance of *different layer's conceptor matrix* on *SST* Corpus, *bert-tiny* models. The layer(s) of CI-BERT that outperform the conceptor post-processing of the same layer(s) are colored in orange.

| Model | SEAT-6 | SEAT-6b | SEAT-7 | SEAT-7b | SEAT-8 | SEAT-8b | Avg. Abs. | |
|---|---|---|---|---|---|---|---|---|
| GPT2 | -0.510 | 0.057 | -0.274 | -0.186 | -0.369 | -0.313 | | 0.285 |
| **(Wordlist Percentile 1.0)** | | | | | | | | |
| + Conceptor-12 (pronouns) | **0.030** | 0.269 | **-0.137** | **0.044** | **-0.129** | **-0.076** | ↓0.171 | **0.114** |
| + Conceptor-12 (extended) | **0.053** | 0.293 | **-0.163** | **0.057** | **-0.114** | **-0.054** | ↓0.163 | 0.122 |
| + Conceptor-12 (propernouns) | 0.713* | 0.430* | **0.029** | 0.222 | **0.045** | 0.196 | ↓0.013 | 0.272 |
| + Conceptor-12 (all) | **0.443*** | 0.333 | **-0.107** | 0.102 | **-0.088** | 0.065 | ↓0.095 | 0.190 |
| + Conceptor-12 (or) | **0.494*** | 0.269 | **0.065** | 0.236 | **-0.044** | 0.308 | ↓0.049 | 0.236 |
| **(Wordlist Percentile 0.9)** | | | | | | | | |
| + Conceptor-12 (pronouns) | **0.030** | 0.269 | **-0.137** | **0.044** | **-0.129** | **-0.076** | ↓0.171 | **0.114** |
| + Conceptor-12 (extended) | **0.312** | 0.448* | -0.352 | **0.032** | **-0.067** | 0.048 | ↓0.075 | 0.210 |
| + Conceptor-12 (propernouns) | 0.713* | 0.430* | **0.029** | 0.222 | **0.045** | 0.196 | ↓0.013 | 0.272 |
| + Conceptor-12 (all) | **0.443*** | 0.333 | **-0.107** | 0.102 | **-0.088** | 0.065 | ↓0.095 | 0.190 |
| + Conceptor-12 (or) | 0.537* | 0.268 | **0.051** | 0.260 | **-0.043** | 0.361 | ↓0.032 | 0.253 |
| **(Wordlist Percentile 0.8)** | | | | | | | | |
| + Conceptor-12 (pronouns) | **0.030** | 0.269 | **-0.137** | **0.044** | **-0.129** | **-0.076** | ↓0.171 | **0.114** |
| + Conceptor-12 (extended) | **0.312** | 0.448* | -0.352 | **0.032** | **-0.067** | 0.048 | ↓0.075 | 0.210 |
| + Conceptor-12 (propernouns) | 0.713* | 0.430* | **0.029** | 0.222 | **0.045** | 0.196 | ↓0.013 | 0.272 |
| + Conceptor-12 (all) | **0.443*** | 0.333 | **-0.107** | 0.102 | **-0.088** | 0.065 | ↓0.095 | 0.190 |
| + Conceptor-12 (or) | 0.537* | 0.268 | **0.051** | 0.260 | **-0.043** | 0.361 | ↓0.032 | 0.253 |
| **(Wordlist Percentile 0.7)** | | | | | | | | |
| + Conceptor-12 (pronouns) | **0.031** | 0.269 | **-0.136** | **0.045** | **-0.128** | **-0.076** | ↓0.171 | **0.114** |
| + Conceptor-12 (extended) | **0.312** | 0.448* | -0.352 | **0.032** | **-0.067** | 0.048 | ↓0.075 | 0.210 |
| + Conceptor-12 (propernouns) | 0.713* | 0.430* | **0.029** | 0.222 | **0.045** | 0.196 | ↓0.013 | 0.272 |
| + Conceptor-12 (all) | **0.443*** | 0.333 | **-0.107** | 0.102 | **-0.088** | 0.065 | ↓0.095 | 0.190 |
| + Conceptor-12 (or) | 0.538* | 0.269 | **0.051** | 0.260 | **-0.043** | 0.361 | ↓0.031 | 0.254 |
| **(Wordlist Percentile 0.6)** | | | | | | | | |
| + Conceptor-12 (pronouns) | **0.031** | 0.269 | **-0.136** | **0.045** | **-0.128** | **-0.076** | ↓0.171 | **0.114** |
| + Conceptor-12 (extended) | **0.304** | 0.449* | -0.381 | **-0.002** | **-0.106** | 0.016 | ↓0.075 | 0.210 |
| + Conceptor-12 (propernouns) | 0.713* | 0.430* | **0.029** | 0.222 | **0.045** | 0.196 | ↓0.013 | 0.272 |
| + Conceptor-12 (all) | **0.443*** | 0.333 | **-0.107** | 0.102 | **-0.088** | 0.065 | ↓0.095 | 0.190 |
| + Conceptor-12 (or) | 0.519* | 0.255 | **0.036** | 0.244 | **-0.054** | 0.348 | ↓0.042 | 0.243 |
| **(Wordlist Percentile 0.5)** | | | | | | | | |
| + Conceptor-12 (pronouns) | **0.031** | 0.269 | **-0.136** | **0.045** | **-0.128** | **-0.076** | ↓0.171 | **0.114** |
| + Conceptor-12 (extended) | **0.269** | 0.411* | -0.258 | **-0.038** | 0.114 | 0.027 | ↓0.099 | 0.186 |
| + Conceptor-12 (propernouns) | 0.713* | 0.430* | **0.029** | 0.222 | **0.045** | 0.196 | ↓0.013 | 0.272 |
| + Conceptor-12 (all) | 0.565* | 0.445* | **-0.110** | 0.125 | **-0.063** | 0.127 | ↓0.046 | 0.239 |
| + Conceptor-12 (or) | **0.478*** | 0.248 | **0.029** | 0.188 | **-0.016** | 0.317 | ↓0.072 | 0.213 |
| **(Wordlist Percentile 0.4)** | | | | | | | | |
| + Conceptor-12 (pronouns) | **0.060** | 0.271 | **-0.130** | **0.045** | **-0.131** | **-0.074** | ↓0.167 | 0.118 |
| + Conceptor-12 (extended) | **0.276** | 0.410* | -0.255 | **-0.038** | 0.110 | 0.027 | ↓0.099 | 0.186 |
| + Conceptor-12 (propernouns) | 0.730* | 0.393* | **0.033** | 0.190 | **0.028** | 0.127 | ↓0.035 | 0.250 |
| + Conceptor-12 (all) | 0.648* | 0.374* | **-0.036** | 0.154 | **0.014** | 0.134 | ↓0.058 | 0.227 |
| + Conceptor-12 (or) | 0.564* | 0.258 | **0.057** | 0.196 | **0.014** | 0.287 | ↓0.056 | 0.229 |
| **(Wordlist Percentile 0.3)** | | | | | | | | |
| + Conceptor-12 (pronouns) | **0.092** | 0.316 | **-0.001** | **0.064** | **-0.035** | **-0.062** | ↓0.190 | **0.095** |
| + Conceptor-12 (extended) | **0.264** | 0.369 | -0.261 | **-0.043** | 0.115 | 0.015 | ↓0.107 | 0.178 |
| + Conceptor-12 (propernouns) | 0.729* | 0.391* | **0.030** | 0.187 | **0.028** | 0.125 | ↓0.037 | 0.248 |
| + Conceptor-12 (all) | 0.682* | 0.397* | **-0.021** | 0.162 | **0.018** | 0.140 | ↓0.048 | 0.237 |
| + Conceptor-12 (or) | 0.545* | 0.214 | **0.072** | 0.198 | **0.006** | 0.279 | ↓0.066 | 0.219 |
| **(Wordlist Percentile 0.2)** | | | | | | | | |
| + Conceptor-12 (pronouns) | **0.094** | 0.316 | **-0.001** | **0.064** | **-0.033** | **-0.062** | ↓0.190 | **0.095** |
| + Conceptor-12 (extended) | **0.249** | 0.356 | -0.238 | **-0.048** | 0.106 | 0.016 | ↓0.116 | 0.169 |
| + Conceptor-12 (propernouns) | 0.699* | 0.403* | **0.042** | 0.176 | **0.017** | 0.092 | ↓0.047 | 0.238 |
| + Conceptor-12 (all) | 0.699* | 0.437* | **0.056** | 0.164 | **0.063** | 0.098 | ↓0.032 | 0.253 |
| + Conceptor-12 (or) | 0.519* | 0.223 | **0.086** | 0.191 | **-0.004** | 0.255 | ↓0.072 | 0.213 |
| **(Wordlist Percentile 0.1)** | | | | | | | | |
| + Conceptor-12 (pronouns) | **0.446*** | 0.599* | **0.035** | 0.064 | 0.064 | -0.037 | ↓0.077 | 0.208 |
| + Conceptor-12 (extended) | **0.497*** | 0.544* | **0.019** | 0.097 | 0.171 | 0.074 | ↓0.051 | 0.234 |
| + Conceptor-12 (propernouns) | 0.753* | 0.495* | **-0.038** | 0.095 | **-0.010** | 0.062 | ↓0.043 | 0.242 |
| + Conceptor-12 (all) | 0.730* | 0.508* | **-0.009** | 0.086 | 0.102 | 0.070 | ↓0.034 | 0.251 |
| + Conceptor-12 (or) | 0.914* | 0.568* | **0.258** | 0.247 | **0.021** | 0.264 | ↑0.094 | 0.379 |

Table 19: SEAT effect size of gender debising. The impact of *different percentiles of wordlist* (using *UMAP* clustering) on *Brown* Corpus, *gpt-2* models. The top-3 best performance is colored in orange.

| Model | CoLA | MNLI | MRPC | QNLI | QQP | RTE | SST | STS-B | WNLI | | Average |
|---|---|---|---|---|---|---|---|---|---|---|---|
| BERT-L | 62.82 | 86.13 | 88.32 | 92.15 | 91.56 | 69.31 | 93.81 | 90.00 | 35.21 | | 78.81 |
| + Conceptor P.P. | 62.34 | 86.16 | 89.44 | 91.71 | 91.59 | 74.73 | 93.58 | 90.06 | 30.17 | ↑0.05 | 78.86 |
| GPT2 | 29.10 | 82.43 | 84.51 | 87.71 | 89.18 | 64.74 | 91.97 | 84.26 | 43.19 | | 73.01 |
| + Conceptor P.P. | 35.47 | 82.39 | 84.08 | 88.30 | 89.12 | 67.15 | 92.09 | 83.67 | 33.80 | ↓0.11 | 72.90 |
| GPT2-L | 12.48 | 88.80 | 82.98 | 80.30 | 87.61 | 51.26 | 81.77 | 78.87 | 46.48 | | 75.84 |
| + Conceptor P.P. | 20.03 | 88.92 | 82.78 | 79.64 | 87.65 | 50.54 | 82.34 | 78.26 | 40.85 | ↑0.04 | 75.89 |
| GPT-J | 59.73 | 82.49 | 87.95 | 87.93 | 87.54 | 75.81 | 94.50 | 88.60 | 39.44 | | 78.22 |
| + Conceptor | 59.48 | 82.89 | 87.69 | 91.56 | 89.90 | 76.17 | 95.07 | 88.81 | 30.99 | ↓0.14 | 78.06 |

Table 20: GLUE validation set results for other LLMs. We use the F1 score for MRPC, the Spearman correlation for STS-B,and Matthew's correlation for CoLA. For all other tasks,we report the accuracy.

| Model | SEAT-6 | SEAT-6b | SEAT-7 | SEAT-7b | SEAT-8 | SEAT-8b | | Avg. Abs. |
|---|---|---|---|---|---|---|---|---|
| BERT | 0.931* | 0.090 | -0.124 | 0.937* | 0.783* | 0.858* | | 0.620 |
| + Gender Conceptor | **0.388** | **-0.078** | -0.292 | **0.179** | **0.594*** | **0.335** | ↓0.309 | **0.311** |
| + Intersected Conceptor | **0.916*** | **0.026** | -0.180 | **0.840*** | **0.749*** | **0.832*** | ↓0.029 | **0.591** |

Table 21: BERT intersectional gender debiasing, where intersected conceptor indicates the conceptor matrix generated by its negated AND operation of gender conceptor matrix and race conceptor matrix

| Model | ABW-1 | ABW-2 | SEAT-3 | SEAT-3b | SEAT-4 | SEAT-5 | SEAT-5b | | Avg. Abs. |
|---|---|---|---|---|---|---|---|---|---|
| BERT | -0.079 | 0.690* | 0.778* | 0.469* | 0.901* | 0.887* | 0.539* | | 0.620 |
| + Race Conceptor | **-0.063** | **0.682*** | 0.803* | **0.209** | 0.949* | 0.946* | **0.390*** | ↓0.043 | **0.577** |
| + Intersected Conceptor | **-0.045** | **0.685*** | 0.799* | **0.361*** | 0.926* | 0.931* | **0.484*** | ↓0.016 | **0.604** |

Table 22: BERT intersectional race debiasing, where intersected conceptor indicates the conceptor matrix generated by its negated AND operation of gender conceptor matrix and race conceptor matrix