# OpenReview forum: "Conceptor-Aided Debiasing of Large Language Models"
_EMNLP/2023/Conference — EMNLP 2023 Main_

### Official Review · Reviewer_cvA6 · 2023-07-28

**Soundness:** 4

**Excitement:**

4: Strong: This paper deepens the understanding of some phenomenon or lowers the barriers to an existing research direction.

**Paper Topic And Main Contributions:**

This paper propose to use conceptor to mitigate bias in language models, specifically, one post-processing-based method and one continuing training-based method. The authors cleverly use the conceptors' logical semantics (AND and OR) to study the effect of combining different bias attribute word lists, and mitigating intersectional bias. The authors also conducted comprehensive experiments to study the impact of different debiasing corpora, bias word lists and their combination and filtering methods, the proposed conceptor methods' performance on different kinds and sizes of language models, as well as the impact on language understanding.

**Questions For The Authors:**

A. You stated that conceptor post-processing achieved SOTA debiasing results in the paper (e.g., lines14-17), but in Table1 , it seems INLP's average absolute SEAT score (0.204) is the lowest. Any comment on this?

**Reasons To Accept:**

The authors proposed two novel methods based on conceptors: one post-processing-based method and one continuing training-based method. Both of them debias the representations in every layer of a transformer and seem to be effective in mitigating bias.

It is novel to use the conceptors' logical semantics (AND and OR) to study the effect of combining different bias attribute word lists, and mitigating intersectional bias. The problems themselves are also of research importance.

The authors conducted extensive experiments and provide many insights on the practice of bias mitigation, such as how to combine different word lists and filter out outliers, the impact of different debiasing corpora, their methods' performance on different kinds and sizes of language models, etc.

**Reasons To Reject:**

I think the authors left out some details for their Algorithm 1 (at least for me). In the original HardDebias [Bolukbasi, etal, 2016][Liang, etal, 2020], the bias subspace is created by the first n (n=1 or 2) principle components of the set of difference vectors between each pair of bias attribute words. But in algorithm 1, it seems the authors simply stack all the word vectors in matrix X, not the difference vectors. I'm wondering why this can mitigate bias. Perhaps the authors could provide more details on the application of conceptors for debiasing, in the related works section or background section.

I'm also a bit confused with the "Conceptor Intervention and Continued Training" method in section 3.2. The authors may want to apply the conceptors to BERT's weights before calculating the loss function and the gradients. But since the word vectors will change after BERT's weights have changed, I'm not sure if the authors re-calculate all the word vectors and the conceptor matrix after every update step of BERT's weight? If so, the method may be computationally expensive.

Some of the main results (such as Table 8-11) are in the appendix. The authors may re-organize the paper and include at least part of these results in the main text for better presentation.

**Reproducibility:**

3: Could reproduce the results with some difficulty. The settings of parameters are underspecified or subjectively determined; the training/evaluation data are not widely available.

**Reviewer Confidence:**

3: Pretty sure, but there's a chance I missed something. Although I have a good feel for this area in general, I did not carefully check the paper's details, e.g., the math, experimental design, or novelty.

---

> ### Author Rebuttal · Authors · 2023-08-28
>
> Thank you for your time and valuable comments.
>
> RE Reject - Conceptor mechanism: As mentioned in Appendix A, the word matrix $X$ is the data collection of the embedded attribute words. Via formula 2, we can compute the $C$–-a soft projection matrix on the linear subspace where the word embeddings $X$ have the highest variance. Once $C$ has been learned, it can be ‘negated’ by subtracting it from the identity matrix and then applied to any word embeddings to shrink the bias directions. Please further note that we do not want to compute the conceptor on the “diff space” as the projection matrix is applied to the original space. In this way the conceptor method is different from the so-called hard-debiasing of Bolukbasi et al. We will add this explanation and move some mechanism introduction from Appendix A to the main body in the camera-ready version.
>
> RE Reject - section 3.2: We find the conceptor matrices by the original BERT’s weights, then use them to continually train the BERT model in a reasonable amount of iterations. We don’t recalculate the word vectors and the conceptor matrix after each iteration, thus it is not computationally expensive.
>
> RE Reject - layout: The main results (e.g. tuned conceptor result, robustness and extendability) are self-contained in the main body (e.g. Table 1-5), and Table 8-11 are the extension of the tuned result in Table 1, and the further proof of the robustness and extendability. However, we understand the confusion of the crowded contents and will adjust the layout when we have an additional page in the camera-ready version, following your suggestion!
>
> RE Question A: What we said is that the conceptor method achieves SoTA debiasing performance *while maintaining the semantic meaning* (line 14-17). As shown in Table 5, INLP drops the GLUE of BERT by 0.99 [1], which means INLP significantly harms the language capacity. In this case, it is suspicious that INLP successfully debiases since the damage of language capacity can also manifest as the decrease of bias in the fairness benchmark. A reductio ad absurdum would be to zero out or randomize all embeddings. This is perfect debiasing; however, has little to no utility. Thus it is critical to find  bias mitigation  methods that do not impair the language capacity in the final target application.
>
> ---
> [1] [An Empirical Survey of the Effectiveness of Debiasing Techniques for Pre-trained Language Models](https://aclanthology.org/2022.acl-long.132) (Meade et al., ACL 2022)

---

### Official Review · Reviewer_ogLa · 2023-08-04

**Soundness:** 3

**Excitement:**

3: Ambivalent: It has merits (e.g., it reports state-of-the-art results, the idea is nice), but there are key weaknesses (e.g., it describes incremental work), and it can significantly benefit from another round of revision. However, I won't object to accepting it if my co-reviewers champion it.

**Paper Topic And Main Contributions:**

The paper focuses on the problem of debiasing existing language models for improving soclal fairness of NLP. The authors challenges the existing conclusion that conceptor negation fails to debias BERT stably. They employ conceptor negation post processing to debias LMS such as GPT while retaining useful semantics and robustness. The authors also explore conceptor-intervened BERT (CIBERT) where they continue training BERT after incorporating conceptors with all BERT layers. With different corpora, bias attribute wordlists and outlier removal criteria they observe different debiasing performance. The results demonstrate that conceptor post-processing outperformsn several SOTA debiasing approaches on both debiasing effectiveness and semantic retention. Proposed CI-BERT outperforms post-processing approach. However, with increased instability and worse semantic retention.

**Reasons To Accept:**

The paper is well written and the proposed approach appears promising given the improvement performance across multiple LMs and on multiple aspects.

**Reasons To Reject:**

As points out by the authors also, the proposed approach is restricted to only English while all recent LMs are all multilingual. Also, the focus has been on debiasing biases in North America and more extensive study would have been more helpful.

**Reproducibility:**

3: Could reproduce the results with some difficulty. The settings of parameters are underspecified or subjectively determined; the training/evaluation data are not widely available.

**Reviewer Confidence:**

1: Not my area, or paper was hard for me to understand. My evaluation is just an educated guess.

**Typos Grammar Style And Presentation Improvements:**

Typos:
1. In line 529, replace “Bert” with “BERT”

---

> ### Author Rebuttal · Authors · 2023-08-28
>
> Thank you for your time and valuable comments. We will fix the mentioned typo in the camera-ready version.
>
> RE Reject #1: We acknowledge and appreciate your point regarding the scope of the recent LMs being multilingual. However, the primary focus of our work was to shed light on the methodology and robustness of the conceptor debiasing technique, and to explore intersectional debiasing, within the context of English LMs. It's worth noting that the specificity of the study (i.e., focusing on English and North American biases) was purposefully chosen to provide a more in-depth analysis of conceptor debiasing pipeline and intersectional biases in this particular context. Broadening the scope to multilingual models would have made the study much more extensive and might have diluted the depth of analysis we intended to provide. That said, we recognize the potential importance of studying such methods in multilingual contexts, and have indeed mentioned it in the limitations section. We believe this offers a direction for future research, where researchers can expand upon our methodology to a multilingual setting.

---

### Official Review · Reviewer_ZbQ9 · 2023-08-11

**Soundness:** 4

**Excitement:**

3: Ambivalent: It has merits (e.g., it reports state-of-the-art results, the idea is nice), but there are key weaknesses (e.g., it describes incremental work), and it can significantly benefit from another round of revision. However, I won't object to accepting it if my co-reviewers champion it.

**Paper Topic And Main Contributions:**

The paper use conceptors to reduce social biases in large language models (LLMs) like BERT and GPT, including two approaches: 1) post-processing to remove bias, maintaining model performance; 2) a new model architecture called CI-BERT that includes bias reduction during training, but with a slight accuracy trade-off. Conceptors effectively handle intersectional bias and maintain robustness. The abstract also highlights the importance of carefully defining the bias subspace for optimal results, involving steps like outlier removal and embedding computation from a cleaner data source.

**Questions For The Authors:**

See reasons to reject
1. What is the difference between Conceptor P.P. (default) and Conceptor P.P. (tuned) in table 1? Did not find any explanation on this.

**Reasons To Accept:**

1. State-of-the-Art De-Biasing performance
2. Bias subspace projection improves the debiasing performance while maintaining the models’ accuracy.
3. The introduction of a new architecture (CI-BERT) that incorporates bias reduction during training shows an innovative and proactive approach to the problem, potentially leading to more effective bias reduction in the long term.
4. Addressing biases in language models is an ethical imperative.

**Reasons To Reject:**

1.  The conceptors method and CI-BERT architecture could introduce complexity and implementation challenges. If the method requires extensive modifications to existing models or intricate procedures, it might hinder its adoption and integration into real-world applications.
2. The focus on constructing a specific bias subspace might lead to overfitting, where the method becomes too tailored to the specific training data and fails to generalize well to new or unseen data.

**Reproducibility:**

4: Could mostly reproduce the results, but there may be some variation because of sample variance or minor variations in their interpretation of the protocol or method.

**Reviewer Confidence:**

3: Pretty sure, but there's a chance I missed something. Although I have a good feel for this area in general, I did not carefully check the paper's details, e.g., the math, experimental design, or novelty.

---

> ### Author Rebuttal · Authors · 2023-08-28
>
> Thank you for your time and valuable comments.
>
> RE Reject #1: The conceptors post-processing method is shown by Table 1 that it can be generalizable to different types and scales of language models (LMs). The post-processing technique is cheap since it doesn’t require any retraining nor finetuning, and can be plugged-in anywhere. Furthermore, we have two settings, as mentioned in line 444-447 and 466-468: (1) *default*: the LM uses the default pipeline setting—Brown corpus, no outlier filtering, and the AND subspace. (2) *tuned*: the LM adopts the strongest pipeline setting after grid-searching different corpora, wordlist percentiles, and subspaces. Per Table 1, even if we use the *default* setting, the conceptor post-processing method can already stably lower the bais. The conceptor continued-training method--which leads to lower bias than the post-processing method--is indeed more computationally expensive. However, the continued training step only requires hundreds of training steps. For example, in the case of bert-base-uncased, it takes less than a half hour on the NVIDIA V100 GPU on Colab. That is, the users can use either the cheap post-processing method under the default setting to get a good debiasing result, or a slightly more expensive (but still the cost is reasonably priced) continued-training method if they want to debias further.
>
> RE Reject #2: Different bias subspaces do lead to different *magnitude* of debiasing performance, but almost all of them debias *successfully*. As said in line 165-168 of the paper, even different choices of bias attribute wordlist would impact the debiasing, and commonly exist in lots of debiasing methods [1]. Therefore, our paper specifically shows the robustness of conceptor methods. For example, the conceptor methods can successfully debias in almost any case no matter which bias subspace, corpora, or wordlist are chosen (Tables 9-11, highlighted in green). In addition, we compute that the overlap of words between SEAT benchmark and the wordlist used for subspace constructed is 0.64%, similar to that of the ‘unseen’ test setting.
>
> RE Q1: See the first response. We will make the definition of *default* and *tuned* more obvious in the table in the camera-ready version of our paper.
>
> ---
> [1] [Bad Seeds: Evaluating Lexical Methods for Bias Measurement](https://aclanthology.org/2021.acl-long.148) (Antoniak & Mimno, ACL-IJCNLP 2021)

---

### Meta-Review · Area_Chair_w4g5 · 2023-09-20

**Recommendation:** 5

**Metareview:**

This paper delineates an approach that employs conceptors for the purpose of debiasing Large Language Models (LLMs). The paper introduces a novel architecture capable of diminishing bias while simultaneously preserving high accuracy. The architecture is characterized as cost-effective and potentially integrable into various architectures, both of which are highly desirable attributes for debiasing methods.

All reviewers unanimously concurred on the novelty of the proposed methods, as well as the rigor and the favorable outcomes of the experiments.

I urge the authors to refine the paper, taking into careful consideration the feedback provided by all the reviewers, with a particular emphasis on addressing the insights from Reviewer 3.

---

### Decision · Program_Chairs · 2023-10-07

**Decision:**

Accept-Main

**Comment:**

This paper delineates an approach that employs conceptors for the purpose of debiasing Large Language Models (LLMs). The paper introduces a novel architecture capable of diminishing bias while simultaneously preserving high accuracy. The architecture is characterized as cost-effective and potentially integrable into various architectures, both of which are highly desirable attributes for debiasing methods.

All reviewers unanimously concurred on the novelty of the proposed methods, as well as the rigor and the favorable outcomes of the experiments.

I urge the authors to refine the paper, taking into careful consideration the feedback provided by all the reviewers, with a particular emphasis on addressing the insights from Reviewer 3.